# Enterprise digital transformation's impact on stock liquidity: A corporate governance perspective

**Hui Liu, Jia Zhu** *, **Huijie Cheng**

School of Business, Zhengzhou University of Aeronautics, Zhengzhou, Henan, China

* zhujia@zua.edu.cn

## Abstract

The innovation in technology and economic growth, which are brought about by digital transformation in enterprises, will inevitably impact their performance in the capital market. Using a sample of Chinese A-share listed companies from 2012 to 2021, this study extensively examines the impact, mechanism, and economic consequences of enterprises digital transformation on stock liquidity. The research reveals that enterprises digital transformation can significantly improve stock liquidity. From the perspective of corporate governance, a further analysis indicates that the digital transformation of enterprises can improve stock liquidity by three mechanisms: easing financing constraints, improving the quality of internal control, and enhancing information disclosure. The results of the heterogeneity analysis indicate that the digital transformation of enterprises, combined with a high level of financial technology, developed financial markets, and policy guidance, has a significantly more significant effect on improving stock liquidity. The analysis of economic consequences reveals that the digital transformation of enterprises can lower the risk of a stock price crash and enhance the accuracy of analysts' forecasts, primarily by improving stock liquidity. This study offers empirical evidence from a micro-mechanism perspective that elucidates the spillover effect of enterprise digital transformation on the capital market. It provides insight into the impact of enterprise digital transformation on stock liquidity and offers theoretical guidance to promote the adoption of enterprise digital transformation across different countries and enhance stock liquidity in the capital market.

## 1 Introduction

In the digital economy, digital technologies, such as artificial intelligence, blockchain, cloud computing, and big data, have rapidly advanced, and digitalization has gradually become a significant breakthrough in restructuring production factors, reshaping economic structures, and transforming competition patterns in economic development. According to studies, companies can improve their business processes and models [1], enhance organizational management [2], promote innovation, and increase enterprise value creation and acquisition [3]. Moreover, various levels of digital transformation can improve market competitiveness and

**Data Availability Statement:** All relevant data are within the paper and its Supporting Information files.

**Funding:** After submitting the manuscript for the first time, we confirmed that "Key Research Project

Plan for Higher Education Institutions of Henan Province(23A630007)" could provide financial support for our study.

**Competing interests:** The authors have declared that no competing interests exist.

sustainable development capabilities by expanding the density of digital technology and improving the depth and breadth of digital technology application. At the same time, enterprises represent the micro-foundation of the capital market. The digital transformation will inevitably impact the functioning of the capital market of enterprises, including price discovery, market policy transmission, information transmission, expectation management, risk mitigation, and other related aspects [4]. Transformative and innovative digital transformation behaviors and the resulting improvements in operational efficiency and economic performance have overwhelming advantages in the capital market [5], offering a wealth of information for the adjustment of capital market resource allocation, but they also present challenges to corporate governance [6]. Existing literature has also verified the impact of digital transformation on the capital market from different perspectives, such as the ability of enterprise digital transformation to improve the capital market environment [7], inhibit the risk of stock price crash [8], and reduce stock price volatility [9]. In conclusion, the external capital market effects of enterprise digital transformation have yet to be fully explored, particularly in terms of in-depth mechanism analysis, which is primarily manifested in the important capital market factor of stock liquidity. Hence, the research findings of this study can contribute to the enrichment of the discussion regarding the influence of internal organizational change on external capital markets.

Stock liquidity typically implies that investors can conduct a large number of stock transactions in a brief time frame without experiencing significant price volatility. Stock with higher liquidity have smaller immediate trading losses and lower investment risk. As the lifeline of the capital market, stock liquidity is one of the important core contents of the stock market [10]. It fulfills the functions of price discovery, information flow, and resource optimization while also reflecting the significance of corporate behavior in the capital market recognition. The liquidity of stocks not only has an impact on investor confidence and the value of the stock market, but it also influences the capital market activities of corporations, the effective allocation of market resources, and the stable development of the real economy. In recent years, relevant academic research has focused on the influencing factors and direct economic consequences of this. Existing literature studies stock liquidity from the perspectives of market uncertainty [11], monetary policy [12], asset liquidity [13], financing constraints [14], corporate governance [15], information disclosure [16], among other perspectives. Digital transformation, as a radical change factor in the traditional production and operation model of enterprises, which has an impact on the value of the company itself, also have an impact on the liquidity of the stock in the capital market? If that's the case, how is it accomplished? The answers to these questions hold significant importance in further exploring the effects of corporate digital transformation on the external capital market and enhancing the sustainable development of listed companies.

From a theoretical perspective, in the context of the digital economy, improving corporate governance is an important way to accelerate corporate transformation and upgrading and enhance corporate sustainable development capabilities.The digital transformation of enterprises can provide new momentum to the production business process, influence the organizational governance structure, resource allocation efficiency, and information disclosure system, and propel corporate governance efficiency and quality, thus affecting stock liquidity. This study reveals the micro-level corporate governance mechanism behind the impact of corporate digital transformation on stock liquidity, addressing the research gap in the study of micro-influencing mechanisms. The findings make it more practical for companies to improve their external market performance through digital transformation.

According to the above-mentioned information, China's capital market is suitable as a research sample for the following reasons: First, compared with developed digital economy

countries, China is the most prominent emerging entity of the digital economy. Moreover, the capital market and enterprises are in the continuous exploration stage of the digital economy, and the generalizability of research conclusions to other countries based on China's capital market is easier. Second, China's large-scale digital economy and the considerable number of enterprises implementing digital transformation can provide sufficient data for this study. Therefore, China offers a favorable environment for observing enterprise digital transformation's impact on the capital market. Finally, this study's research conclusions aim to provide experience and inspiration for other countries to examine the relationship between enterprise digital transformation and capital markets.

Therefore, this study investigates the impact, mechanism, and economic consequences of enterprise digital transformation on stock liquidity, considering financing constraints, internal control quality, and information disclosure level within the framework of corporate governance. This study holds significant theoretical and practical value. This study considers Chinese A-share listed companies from 2012–2021 as samples to examine digital transformation's impact on stock liquidity. The expected research objectives are as follows: First, it reveals corporate digital transformation's role in improving stock liquidity and supplements the literature on corporate digital transformation and capital markets from the corporate governance standpoint. Second, this study empirically tests the corporate digital transformation mechanism to enhance stock liquidity from three aspects of corporate governance—financing constraints, internal control, and information disclosure—and discloses the mechanism path between corporate digital transformation and stock liquidity to a certain extent. Third, this study analyzes the heterogeneity of digital transformation's impact on stock liquidity from the perspective of the fintech and financial market development levels and the policy guidance environment of enterprises. Fourth, it confirms that enterprise digital transformation can effectively minimize stock price crash risk and improve the quality of analysts' forecasts after enhancing stock liquidity. In sum, this study extends the research on the economic consequences of enterprise digital transformation from the enterprise level to the performance level of the capital market, offering a novel perspective to examine the factors affecting stock liquidity and enriching the interaction between the digital economy and capital market.

## 2 Theoretical analysis and research hypotheses

### 2.1 Digital transformation and stock liquidity

Stock liquidity is a core factor that reflects the stock market's operating efficiency. It can effectively enhance pricing efficiency's information effect and production efficiency's governance effect, alleviate information asymmetry, reduce transaction costs and market risks, optimize business decisions, improve the capital allocation efficiency of enterprises, and increase corporate value [17]. However, problems such as internal and external information asymmetry and management opportunism caused by agency conflicts have become predominant reasons for inhibiting corporate stock liquidity [18]. Modern corporate governance theory holds that effective corporate governance can restrain short-term management behaviors and enhance management mechanisms and information transparency, thus improving stock liquidity [19]. On the one hand, a high-quality governance environment can expand corporate financing channels, market activity, and recognition, optimize capital structures, balance stock structures and leverage levels, reduce stock costs and information asymmetry, ease financing constraints, and improve stock liquidity [20–22]. On the other hand, efficient corporate governance can enhance the quality of risk disclosures, accounting information, and corporate earnings, improve internal governance mechanisms, product market competition and innovation ability, sustainable development ability, and comprehensive strength of enterprises, and reduce

investment and debt default risks, thus improving stock liquidity [23]. Enterprise digital transformation is an essential phase in the digital economy era. The deep integration of enterprise total factors and digital technologies can foster the reorganization of production resources and business process reengineering, enhance corporate governance capabilities, corporate value, and internal governance mechanisms, and optimize the corporate governance environment [24]. Notably, this is achieved through the following ways. First, digital transformation can tackle the problem of financing constraints in corporate governance, create internal and external supervisory mechanisms by reducing information asymmetry, acquire support from external investors, creditors and the government, and promote corporate financing [25,26]; Second, digitalization can significantly enhance the quality of internal control of enterprises. Through internal and external cooperation, enterprises can optimize factor allocation, lessen agency problems, strengthen the internal environment, information communication efficiency, and risk prevention capacity, and thus improve the internal control level of enterprises [27]. Finally, the enterprise digital transformation can effectually improve the information disclosure level. Enterprises enrich the information transmission channels through digital technology, improve the ability of information mining and processing, strengthen the internal information disclosure system, and ensure information authenticity, thus enhancing the quality of information disclosure [28,29]. Therefore, from the corporate governance perspective, changes in information efficiency and governance mechanisms driven by the enterprise digital transformation can be reflected in the capital market and have an enhancement effect on stock liquidity [30]. Based on the above analysis, the following hypothesis is proposed:

H1: Digital transformation of enterprises can improve stock liquidity.

## 2.2 The impact mechanism of digital transformation on stock liquidity

**2.2.1 Digital transformation, financing constraints, and stock liquidity.** According to the corporate governance theory, information asymmetry inside and outside the enterprise leads to high adverse selection costs and inhibits the stock market's performance. However, by expanding financing channels and optimizing capital structure and share capital structure, enterprises can enhance information transparency and market recognition and ease financing constraints, which are conducive to improving stock liquidity. Current research suggests that enterprise digital transformation can effectively reduce financing constraints [31]. First, the digital transformation of enterprises improves corporate information transparency, operational efficiency, and quality of risk control. It reduces the information and decision-making costs and valuation risk of stakeholders, makes obtaining financial support from external shareholders and creditors easier, and curtails financing constraints. Second, enterprise digital transformation can help the whole supply chain regulate the value creation mode according to demand, shatter spatial boundaries, strengthen supply chain cooperation, enhance commercial credit support from suppliers and customers, and reduce financing constraints [32,33]. Third, the innovative behavior of enterprise digital transformation helps attain direct or indirect financial subsidies from the government. The certification effect of government subsidies helps transmit positive signals, obtain more social financial support, and ease the enterprises' financing constraints [34,35]. Fourth, the digital transformation of enterprises will transfer more high-quality information to the market, improve the information transparency and information disclosure quality of enterprises, reduce the information asymmetry between enterprises and creditors and potential investors, reduce the financing costs and obstacles of enterprises, and ease the financing constraints [36]. At the same time, the enterprise digital transformation to build digital management system greatly improve the enterprise internal

information processing ability and information transparency, reduce enterprise internal information asymmetry, provide information support for management decision, enhance enterprise financial decision-making ability and financial operation efficiency, is the enterprise under the limited financial resources to maximize resource use, ease the financing constraints (Ren et al.,2023). Therefore, digital transformation lays the groundwork for enterprises to curtail financing constraints. The continuous reduction of corporate financing constraints enhances stock price information efficiency and investment efficiency, boosts market depth and corporate goodwill, and reduces bid-ask spreads and adverse selection costs, thus improving stock liquidity. Therefore, the following hypothesis is proposed:

H2: Digital transformation can improve stock liquidity by reducing financing constraints.

**2.2.2 Digital transformation, internal control, and stock liquidity.** According to the agency theory, agency conflict among corporate stakeholders may lead to adverse selection and the moral hazard of management, raise corporate governance costs and business risks, and restrict stock liquidity [37]. Ideal corporate governance can create an efficient internal control decision-making mechanism and internal and external supervisory mode. Additionally, it can limit the short-term behavior of management, safeguard shareholders' rights and interests, enhance internal and external communication efficiency, increase stock returns and corporate value, reduce stock price fluctuations, and guarantee stock liquidity [38,39]. Enterprise digital transformation can efficiently improve the quality of internal control and the management checks and balances mechanism, optimize the internal governance framework through digital technology, promote governance decisions and behavior changes, and enhance the corporate governance level [40]. This is accomplished through the following ways. First, emerging digital technologies furnish enterprises with accurate governance information, strengthen risk early warning, assessment, and disposal capabilities, optimize factor allocation, improve production and operation quality and efficiency, enhance profitability quality and growth capacity, reduce agency costs, shatter traditional governance boundaries, create multi-departmental collaboration, and foster the improvement of internal control [41,42]. Second, digital transformation encourages the participation of external stakeholders in governance, augments internal and external mutual supervision and checks and balances, helps integrate and mine various governance data, creates a network governance environment, and fosters scientific governance decision-making and effective operation of corporate governance [43]. Third, the digital management system built by the enterprise promotes the information exchange and sharing within the enterprise, reduces the information asymmetry within the enterprise, strengthens the internal information communication through improving the internal information transparency of the enterprise, and improves the internal control environment of the enterprise [44]. At the same time, the digital transformation of enterprises brings a network flat organizational framework, which further improves the information transparency and information disclosure quality within the organization, makes the management fully understand the decision-making risk, reduces the herd effect of decision-making, and improves the quality of internal control [45]. Therefore, digital transformation improves the enterprises' quality of internal control, creates internal and external coordination supervision and governance mechanisms, overcomes the "digital divide", forms an efficient internal control system, regulates internal and external information synchronization, mitigates agent conflicts, transmits good market signals, stimulates the stock market activity, and thus enhances stock liquidity. Therefore, the following hypothesis is proposed:

H3: Digital transformation can improve stock liquidity by improving the quality of internal control.

**2.2.3 Digital transformation, information disclosure, and stock liquidity.** Information transfer theory holds that information drives the capital market, and the market's stable operation depends on effective information disclosure. Due to adverse selection costs, investors often seek better trading prices to compensate for trading losses caused by the information disadvantage, and the transaction costs resulting from information disclosure will impact stock liquidity [46]. Rich information disclosure can efficiently reflect the intrinsic value of enterprises, strengthen the correlation between accounting information and economic income, diminish investors' valuation risks and information processing costs, and evade investors' value misjudgment [47]. Thus, higher information disclosure can effectively decrease the information asymmetry among stakeholders, enhance investors' market attention and trading confidence, attract more investor participation, and thus foster the improvement of stock liquidity [48,49]. For enterprises with poor information disclosure, investors lack the basis for value judgment and cannot efficiently assess enterprise value, increasing the likelihood of internal transactions, agent conflicts, and other difficulties and reducing investors' willingness to hold, which leads to diminished stock liquidity [50,51]. Enterprise digital transformation disrupts the "information island", extends the information search channel through digital technology, enhances the scope and quality of enterprise information disclosure, and alleviates the problem of stakeholder information bias and adverse selection. On the one hand, digital technology transforms multi-dimensional non-standardized and unstructured information in enterprise production management into standardized information, improves non-financial information disclosure such as business model, governance structure, and risk management, expands the width and depth of information disclosure, and enhances information transparency [52]. On the other hand, digital technology augments the enterprise information processing ability, ensures the timeliness and accuracy of information disclosure, and improves its readability and economic value. Furthermore, it increases the disclosure of enterprise characteristic information and investor preference information, reduces the possibility of enterprise concealment and selective disclosure, and enhances the overall quality of enterprise information disclosure [53]. Therefore, enterprises improve information disclosure through digital transformation, reduce internal and external information asymmetry, develop stakeholders' information communication efficiency and value judgment ability, strengthen investors' attention and trading willingness, and thus enhance stock liquidity. Therefore, the following hypothesis is proposed:

H4: Digital transformation can improve stock liquidity by increasing the level of information disclosure.

The following **Fig 1** is the research framework of this study.

## 3 Research design

### 3.1 Data and sample

The concept of "digital transformation" was first proposed by IBM in 2012. To ensure continuity and an adequate sample size during the research phase, and according to the empirical testing practices at home and abroad, and considering the huge impact of the novel coronavirus on Chinese enterprises may lead to abnormal sample data, the node of the study period is determined as 2021 and the study period as 10 years, namely 2012–2021. Building on previous analysis, China stands as the largest emerging player in the digital economy, with capital markets and enterprises continuing to explore this new frontier. The research conclusions derived from China's capital market are more easily applicable to other countries. At the same time, the scale of China's digital economy is vast, and numerous enterprises among China's A-share

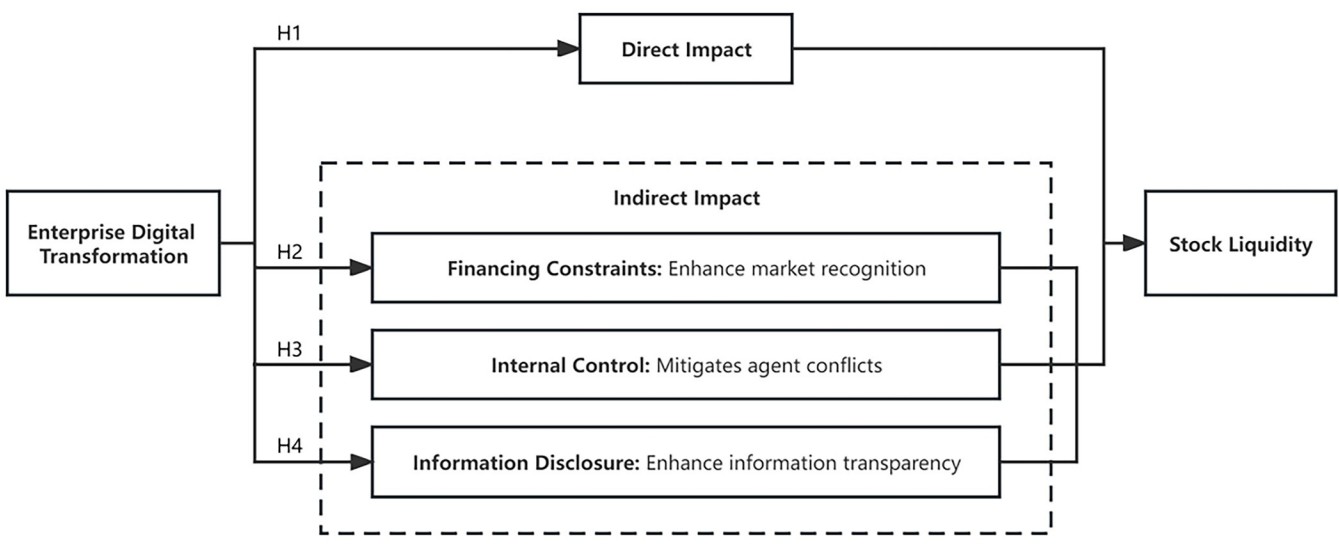

**Fig 1. Research framework.**

listed companies are implementing digital transformation, providing ample data for this study. We have selected China's A-share listed companies as our research object. At the same time, the scale of China's digital economy is relatively large, and many enterprises in China's A-share listed companies have undergone digital transformation, providing sufficient data for this study. Therefore, we have decided upon China's A-share listed companies as our research subject. This study selects enterprises that have implemented digital transformation among China's A-share listed companies from 2012 to 2021 as the research sample, and processes the data as follows: (1) Excluding ST and *ST samples and samples that have been forced to delist during the sample period to avoid abnormal fluctuations in data caused by continuous losses of enterprises; (2) Excluding data anomalies and missing data samples to avoid empirical test bias caused by missing data; (3) Excluding samples from the financial and insurance industries to avoid anomalies in empirical testing caused by the special financial situation of the financial and insurance industries. Simultaneously, to reduce the impact of extreme values in the sample data on the empirical test results, 1% and 99% Winsorize tail processing were adopted for the selected variable sample data. The original data of enterprise digital transformation, namely the annual report of listed companies, comes from Cninfo. Raw data of stock liquidity, Fund guarantee, Enterprise growth, Ownership concentration, Business investment opportunity, Zeros index, Roll index, Bid-ask Spread, Afq, and Spcr, from CSMAR database. Enterprise scale, Asset liquidity, Profitability, Financial leverage, and Stock return, from Wind Database Terminal. This study utilizes the Stata17.0 software for processing data and conducting empirical analysis. Among them, the digital transformation data of enterprises may be affected by the factors of enterprise characteristics and operating environment and lead to the sample deviation; the Amihud index calculation of stock liquidity may be affected by the enterprise size and the stock holding period; In the selection process of control variables, this study tried to select the influencing factors related to the study subjects, and could not choose all the influencing factors as the control variables. Therefore, in order to solve these potential sample bias and limitation problems, this study should focus on these issues in the endogeneity and robustness test.

### 3.2 Variable setting and description

**3.2.1 Explained variable: Stock liquidity.** Stock liquidity (Liquidity) is the market's ability to trade assets at a reasonable price, reflecting the quality of the capital market's operations. Existing research on liquidity primarily constructs liquidity indicators from the perspectives of price, trading volume, and combination of price and volume. According to Chiang and Zheng (2015), Le and Gregoriou et al. (2020), the combined measure of price-volume illiquidity takes into account the impact of stock trading volume and price, accurately reflecting the relationship between trading volume and trading price. This measure is considered to be objective and precise in assessing stock liquidity. Therefore, the illiquidity measure is chosen in this study to gauge stock liquidity. We utilize Amihud and Mendelson's [54] Formula (1) on liquidity to compute the stock's illiquidity index (ILLIQ).

$$\mathbf{ILLIQ_{i,t}} = \frac{\mathbf{1}}{\mathbf{D_{i,t}}} \sum\nolimits_{\mathbf{d=1}}^{\mathbf{D_{i,t}}} \sqrt{\frac{|\mathbf{r_{i,t,d}}|}{\mathbf{V_{i,t,d}}}} \tag{1}$$

where, $\mathbf{D_{i,t}}$ denotes the annual trading days of the enterprise stock; $\mathbf{V_{i,t,d}}$ denotes the stock transaction amount of the enterprise on the trading day; $\mathbf{r_{i,t,d}}$ denotes the rate of return of cash dividends reinvested in the business day. The illiquidity index (ILLIQ) reflects the impact of the amount of each unit of transaction on the stock price. The larger the value, the lower the stock's liquidity. This study considers the time lag and effectiveness of enterprise digital transformation's impact on the capital market, and proposes a one-period increase in illiquidity [55]. To streamline the subsequent empirical research, Liquidity in this is calculated as the reciprocal of ILLIQ. This relationship indicates that a higher value of the illiquidity indicator corresponds to a higher level of stock liquidity, as illustrated in Formula (2).

$$\mathbf{Liquidity_{i,t}} = -\mathbf{ILLIQ_{i,t+1}} \tag{2}$$

**3.2.2 Explanatory variable: Digital transformation.** In the literature, using the text analysis of listed companies' annual reports to measure enterprise digital transformation (DIG) is a common practice. This study employs text analysis as a measurement method to mirror the primary approaches of enterprise digital transformation and evaluate the impacts of digital transformation on external capital markets and internal business conditions. Referring to Jiang et al. (2022), Fang et al. (2023) and Chen et al. (2023) [56], the specific construction method of text analysis for enterprise digital transformation in this study is as follows: Firstly, we have referred to existing literature, policy documents, and research reports such as the "14th Five-Year Plan for Digital Economic Development" and the "Global Digital Economy White Paper 2022", to identify specific keywords related to enterprise digital transformation. We have then compiled a digital transformation feature word bank, which includes 99 relevant word samples spanning technologies like artificial intelligence, big data, and machine learning [57]. Secondly, the texts from the sections of "Management Discussion and Analysis" and "Business Discussion and Analysis" in the annual reports of listed companies were identified as the sources for enterprise digital transformation. The annual reports of listed companies were downloaded from Juchao Information's website using Python, and the text analysis content was then extracted. Thirdly, the Jieba module of Python is employed to analyze text for word segmentation, which is then matched with a digital feature word library to compute word frequency. Fourthly, the quantitative indicator DIG for enterprise digital transformation is defined as the sum of the frequencies of all digital feature words.

**3.2.3 Control variables.** In order to improve this study's scientific nature, referring to Liu et al. (2023) and Wu et al. (2023) [58], this study governs external market and internal

**Table 1. Variable description and definition.**

| Variable Type | Variable Name | Variable Symbol | Variable Definition |
|---|---|---|---|
| Explained variable | Stock liquidity | Liquidity | Take the opposite number in accordance with the Amihud method; the greater the value, the higher the stock's liquidity. |
| Explanatory variable | Enterprise digitization | DIG | Word frequency statistics of enterprise digital feature words based on text analysis. |
| Control variable | Enterprise scale | Size | Take the natural log of total assets. |
| | Asset liquidity | Cur | Current assets ratio: end-of-term current assets / ending current liabilities. |
| | Profitability | Roe | Roe: Net profit / shareholders' equity balanc. |
| | Financial leverage | Lev | Asset-liability ratio: Total liabilities / Total assets. |
| | Fund guarantee | Netcash | Net cash flow per share: The ratio of the inflow of cash and cash equivalents of the Company minus the outflow balance (net inflow or net expenditure) to the total share capital of the Company. |
| | Enterprise growth | Growth | Revenue growth rate: (Amount of operating income current year-amount of previous year of operating income) / amount of operating income previous year. |
| | Earnings per share | Eps | Earnings per share: The ratio of after-tax profit to the total number of shares |
| | Ownership concentration | Top10 | Share of top 10 shareholders: The sum of the shareholding ratio of the top ten shareholders. |
| | Business investment opportunity | Tobin Q | Tobin Q: Total market value / assets. |

operation variables, which can potentially impact stock liquidity in the model. It includes enterprise size (Size), financial leverage (Lev), profitability (Roe), asset liquidity (Cur), capital security (Netcash), enterprise growth (Growth), Earnings per share (Eps), stock concentration (Top10), and enterprise investment opportunities (TobinQ). **Table** 1 presents the description and definition of each variable.

## 3.3 Model design

To examine the impact of enterprise digital transformation on stock liquidity, we believe that stock liquidity could be influenced by numerous factors in our study. This study employs multiple linear regression models and panel data models to calculate estimated coefficients, which measure the impact of independent variables (enterprise digital transformation) on dependent variables (stock liquidity), while also conducting tests on the hypotheses of this study. This study establishes the following model and employs Stata for empirical testing:

$$\mathbf{Liquidity_{i,t}} = \boldsymbol{\alpha}_0 + \boldsymbol{\alpha}_1 \mathbf{DIG_{i,t}} + \boldsymbol{\alpha}_2 \mathbf{Controls_{i,t}} + \sum \mathbf{Firm} + \sum \mathbf{Year} + \boldsymbol{\varepsilon}_{i,t} \quad (3)$$

Liquidity is the explained variable, representing stock liquidity; DIG is the explanatory variable, representing enterprise digital transformation; Controls is the control variable, and $\boldsymbol{\varepsilon}_{\mathbf{i,t}}$ is the error term. The main estimated parameter $\boldsymbol{\alpha}_1$ represents the digital transformation's impact on stock liquidity. To increase the robustness of the empirical test results, the benchmark regression model controlled the individual fixed effect (Firm) and the time fixed effect (Year) and adopted the robust standard misestimate regression model.

## 4 Empirical test results and analysis

### 4.1 Descriptive statistics of main variables

The mean value of stock liquidity of the explained variable is -14.34, the standard deviation is 7.56, the minimum value is -42.22, and the maximum value is -0.23, indicating significant differences in stock liquidity among different enterprises. This aligns with the characteristics of

**Table 2. Descriptive statistics.**

| Variable | Obs | Mean | Standard Deviation | Minimum | Maximum |
|---|---|---|---|---|---|
| Liquidity | 26,526 | -14.34 | 7.56 | -42.22 | -0.23 |
| DIG | 26,526 | 40.25 | 61.83 | 0 | 435 |
| Size | 26,526 | 22.28 | 1.29 | 19.55 | 26.52 |
| Cur | 26,526 | 56.72 | 20.09 | 7.89 | 96.56 |
| Roe | 26,526 | 5.44 | 14.24 | -125.45 | 33.41 |
| Lev | 26,526 | 42.33 | 20.18 | 4.00 | 90.60 |
| Growth | 26,526 | 14.78 | 33.10 | -65.76 | 280.25 |
| Netcash | 26,526 | 0.08 | 0.82 | -3.96 | 4.70 |
| Eps | 26,526 | 0.38 | 0.61 | -1.88 | 5.04 |
| Top10 | 26,526 | 58.18 | 14.99 | 21.77 | 90.9 |
| Tobin Q | 26,526 | 2.75 | 2.13 | 0.83 | 23.21 |

the capital market performance of various enterprises. The mean value of the explanatory variable digital transformation (DIG) is 40.25, the standard deviation is 61.83, the minimum value is 0, and the maximum value is 435. This indicates significant differences in the degree of digital transformation among enterprises; the sample data has random variability, and some enterprises have not achieved digital transformation. **Table** 2 presents the descriptive statistics of the control variables that are also consistent with the characteristics of economic firms.

## 4.2 Baseline regression analysis

**Table** 3 presents the benchmark regression test results of digital transformation and stock liquidity. Column (1) reports the benchmark regression results without control variables. The results suggest that the estimated coefficient of enterprise digital transformation is significantly positive at a 1% confidence level, indicating that enterprise digital transformation has an enhancing effect on stock liquidity. Column (2) lists the test result when adding some control variables, and Column (3) lists the test result when adding all control variables. The results indicate that under the scenario of gradually adding control variables, the digital transformation of enterprises has an enhancing effect on stock liquidity at the 1% significance level; that is, enterprises can strengthen the link between enterprises and the market through digital transformation. This improves the performance of enterprises in the capital market, thereby enhancing stock liquidity. Thus, H1 is supported.

## 4.3 Robustness test

Homogeneity index substitution test. To guarantee the reliability of the regression test outcomes, this study performed homogenous index substitution tests on indicators of enterprise digital transformation and stock liquidity.

**4.3.1 Replace the explained variable.** To validate the dependability of the explanatory variable, the method of measuring stock liquidity, in this study, we employed the Zeros Index, Roll Index, and Bid-ask Spread (Bas) to replace the Amihud index in the benchmark model for regression testing in a robustness test (Ding and Hou, 2015). According to Brogaard et al. (2017) [59] and Fang et al. (2014) [60], the research practice measures the stock liquidity by using the ratio of the difference between the sell price and the purchase price and the midpoint of the sell and the purchase price. The larger the Zeros Index, Roll Index, and Bid-ask Spread (Bas) are, the lesser the stock liquidity will be (Lesmond et al., 1999; Roll, 1984) [61]. The regression results of replacing the explained variables from the Zeros Index, Roll Index, and

**Table 3. Baseline regression test results.**

| VARIABLES | (1) Liquidity | (2) Liquidity | (3) Liquidity |
|---|---|---|---|
| DIG | 0.106*** | 0.0490*** | 0.0367*** |
| | (7.053) | (3.645) | (2.998) |
| Size | | 2.324*** | 2.962*** |
| | | (17.73) | (22.28) |
| Cur | | 0.0145*** | 0.0236*** |
| | | (3.146) | (5.305) |
| Roe | | 0.0362*** | 0.0117*** |
| | | (11.19) | (2.831) |
| Lev | | -0.0243*** | -0.0339*** |
| | | (-5.037) | (-7.237) |
| Netcash | | 0.0490*** | -0.0241 |
| | | (3.645) | (-0.703) |
| Growth | | | 0.00505*** |
| | | | (4.973) |
| Eps | | | 0.621*** |
| | | | (5.212) |
| Top10 | | | -0.0803*** |
| | | | (-13.78) |
| Tobin Q | | | 0.696*** |
| | | | (24.95) |
| Constant | -22.50*** | -72.87*** | -83.59*** |
| | (-169.6) | (-25.90) | (-29.46) |
| Firm | Yes | Yes | Yes |
| Year | Yes | Yes | Yes |
| Observations | 26,526 | 26,526 | 26,526 |
| R-squared | 0.342 | 0.377 | 0.414 |

Bid-ask Spread are presented in Columns (1), (2) and (3) of **Table 4**. The estimated coefficients are significant at the 1% and 10% confidence levels, respectively, whereas the baseline regression results have not changed significantly. Thus, the core conclusions of this study are still valid.

**Table 4. Robustness test results.**

| VARIABLES | (1) Zeros | (2) Roll | (3) Bas | (4) Liquidity | (5) Liquidity |
|---|---|---|---|---|---|
| DIG | -0.218*** | -0.0496* | -1.096*** | | |
| | (-4.605) | (-1.706) | (-7.069) | | |
| DIG1 | | | | 0.441* | |
| | | | | (1.813) | |
| DIG2 | | | | | 0.550*** |
| | | | | | (3.325) |
| Controls | Yes | Yes | Yes | Yes | Yes |
| Firm | Yes | Yes | Yes | Yes | Yes |
| Year | Yes | Yes | Yes | Yes | Yes |
| Observations | 29,812 | 29,812 | 29,698 | 29,865 | 29,845 |
| R-squared | 0.224 | 0.373 | 0.295 | 0.394 | 0.394 |

**4.3.2 Replace the explanatory variables.** In order to validate the reliability of the explanatory variable, specifically the measurement method for enterprise digital transformation, the digital feature word library construction method and the digital feature word proportion measurement method of Wu et al. (2021) were employed in this study's robustness test, replacing the reference Zhao et al. (2021) enterprise digital transformation measurement method in the benchmark regression. First, Wu et al. (2021) used the construction method of digital feature words to re-measure the degree of enterprise digital transformation and replace the explanatory variable (DIG1). Second, to avoid the influence of the text length of listed companies on the word frequency of digital feature words, the ratio of the word frequency of digital feature words to the total number of words in the full text is used as the replacement explanatory variable (DIG2). Columns (4) and (5) of **Table** 4 present the regression results of replacing the explanatory variables, DIG1 and DIG2. The two digital transformation replacement variables improve stock liquidity at the significance level of 10% and 1%, respectively, indicating that the core conclusions of this study remain robust.

## 4.4 Endogeneity test

**4.4.1 Reverse causation problem.** There may be some endogenous interference of reverse causation between explanatory variables and explained variables; that is, enterprises with good stock liquidity will promote enterprises to carry out digital transformation. Therefore, this study uses two methods to eliminate the possible endogenous interference: explanatory variable lag and instrumental variable.

Lag test of explanatory variables. Given that the economic impacts of digital transformation on the capital market may exhibit a time lag, this study aims to eliminate potential reverse causality issues by testing the core explanatory variable of digital transformation (DIG) with a 1–3 period lag (Yonghong et al., 2023) The inspection results are presented in columns (1), (2), and (3) of **Table** 5. The results indicate that the estimated coefficient of enterprise digital transformation is significantly positive at the 5% confidence level, suggesting that the central finding of this study is robust and trustworthy.

Instrumental variable. This article refers to Peng et al. (2022) and Zhai et al. (2022) [62] in addressing the potential reverse causality issue resulting from unobserved variables, by employing the instrumental variable two-stage least squares method. This method introduces instrumental variables of the explanatory variable (enterprise digital transformation) to verify that the explained variable is only related to the explanatory variable, and is not related to any other unobserved factors that affect the explained variable except for the explanatory variable, thereby eliminating endogenous interference caused by reverse causality. It also refers to the construction method of instrumental variables proposed by Lewbel (1997) [63] and Hua et al. (2023). Mean value of firm digitalization by industry and region as instrumental variable (DIG.IV). Columns (4) and (5) of **Table** 5 present the test results. Among them, in the first stage regression, the estimated coefficient of DIG.IV is significantly positive at a 1% confidence level, suggesting that the instrumental variables are correlated with the endogenous variables. In the second stage regression, the Kleibergen-Paap rk LM statistic was significant at a 1% confidence level, thus rejecting the null hypothesis for insufficient identification of instrumental variables. Kleibergen–Paap rk Wald F statistic was much larger than the Stock-Yogo critical value 16.38. The critical value of the F test at the 10% significance level rejects the null hypothesis of the weak instrumental variable. Therefore, this study's benchmark regression results show no endogenous interference is caused by reverse causality.

Table 5. Endogenous reverse causality test results.

| VARIABLES | (1) Liquidity | (2) Liquidity | (3) Liquidity | (4) DIG | (5) Liquidity |
|---|---|---|---|---|---|
| L1.DIG | 0.257** | | | | |
| | (2.075) | | | | |
| L2.DIG | | 0.321** | | | |
| | | (2.495) | | | |
| L3.DIG | | | 0.319*** | | |
| | | | (2.632) | | |
| DIG.IV | | | | 0.987*** | |
| | | | | (73.88) | |
| DIG | | | | | 0.355*** |
| | | | | | (4.567) |
| Kleibergen–Paap rk LM statistic | | | | | 1926.039*** |
| Kleibergen–Paap rk Wald F statistic | | | | | 5457.796[16.38] |
| Controls | Yes | Yes | Yes | Yes | Yes |
| Firm | Yes | Yes | Yes | Yes | Yes |
| Year | Yes | Yes | Yes | Yes | Yes |
| Observations | 21,819 | 18,326 | 15,210 | 26,526 | 26,526 |
| R-squared | 0.283 | 0.225 | 0.234 | 0.494 | 0.494 |

**4.4.2 Missing variables problem.** In this study, endogenous interference may result from missing variables in the baseline regression. Therefore, to eliminate potential endogenous issues arising from sample omission bias in the research process, this study employs propensity score matching (PSM).

This study employs propensity score matching (PSM) to estimate the net effect of digital transformation on stock liquidity [64], effectively circumventing sample selection issues.This research method avoids the problem of sample selection bias by controlling the influencing factors in the study to correct the estimation bias caused by omitted variables. This method minimizes the endogeneity problem caused by estimation bias by allowing the control group and experimental group to compare the estimated results under similar circumstances.This study divided the control group and the experimental group based on the digital transformation of enterprises is greater than the average valu, select the control variables mentioned in the previous study as covariates, and employed nearest neighbor matching, radius matching, and kernel matching to conduct a with-replacement matching. A balance test was also performed. A regression test was conducted on the samples processed by PSM, and Columns (1), (2), and (3) of **Table** 6 presents the results. The estimated coefficients of enterprise digital transformation were all significantly positive, at least at a 1% confidence level. It implies that the baseline regression results were not affected by the intrinsic interference of the missing variables. Thus, this study's core conclusions remain robust.

## 5 Mechanism analysis

Furthermore, to fully understand the relationship between digitalization and stock liquidity, an empirical test on the mechanism existing between the two is carried out. This study selects financing constraints, internal control, and information disclosure as intermediary mechanisms to test. In order to uncover the mechanism behind the influence of corporate digital transformation on stock liquidity, this study makes reference to Baron and Kenny (1986) [65],

**Table 6. Endogenous omitted variable test results.**

| | (1) | (2) | (3) |
|---|---|---|---|
| | **Nearest neighbor matching** | **Radius matching** | **Nuclear matching** |
| **VARIABLES** | **Liquidity** | **Liquidity** | **Liquidity** |
| DIG | 0.356*** | 0.322*** | 0.360*** |
| | (4.430) | (4.456) | (3.034) |
| Controls | Yes | Yes | Yes |
| Firm | Yes | Yes | Yes |
| Year | Yes | Yes | Yes |
| Observations | 23,556 | 26,515 | 12,981 |
| R-squared | 0.426 | 0.426 | 0.422 |

Wen et al. (2014). This method of testing can concisely and effectively explain the mechanism of mediating variables, not only in effectively reducing experimental error rates, but also in improving empirical testing efficiency. Causal steps approach is employed to examine the relationships between corporate digital transformation, financing constraints, internal control, and information disclosure on one hand, and stock liquidity on the other. Following the benchmark model (3), a mediating effect model was constructed, which was achieved by:

$$\text{Liquidity}_{i,t} = \alpha_0 + \alpha_1 \text{DIG}_{i,t} + \alpha_2 \text{Controls}_{i,t} + \sum \text{Firm} + \sum \text{Year} + \varepsilon_{i,t} \quad (4)$$

$$M_{i,t} = b_0 + b_1 \text{DIG}_{i,t} + b_2 \text{Controls}_{i,t} + \sum \text{Firm} + \sum \text{Year} + \varepsilon_{i,t} \quad (5)$$

$$\text{Liquidity}_{i,t} = c_0 + c_1 \text{DIG}_{i,t} + c_2 M_{i,t} + c_3 \text{Controls}_{i,t} + \sum \text{Firm} + \sum \text{Year} + \varepsilon_{i,t} \quad (6)$$

where M is the intermediate variable financing constraint (FC), internal control (IC), and information disclosure (ID). Other variable definitions are consistent with the baseline regression model(3). This study refers to the SA index constructed by Hadlock and Pierce (2010) [66] to measure enterprise financing constraints (FC) for testing, aiming to reveal the effect of enterprise digital transformation on expanding enterprise financing channels to alleviate financing constraints. This study also uses the Dibo internal control index to measure internal control (IC), aiming to reveal the effect of enterprise digital transformation to improve the quality of internal control. Finally, this study measures the level of information disclosure (ID) by referring to the research method of Bharath and Pasquariello (2009) [67], aiming to reveal the effect of enterprise digital transformation on improving the information disclosure level.

The results of the causal steps approach are presented in **Table 7**. The initial empirical test involves validating the correlation between enterprise digital transformation and stock liquidity, which serves as the research hypothesis H1 of this study. The previous study has confirmed that enterprise digital transformation can significantly enhance stock liquidity, thus it will not be repeated.

The second step in empirical testing is to validate the correlation between enterprise digital transformation and mediating variables. The empirical test results are presented in columns (1), (3), and (5) of **Table 7**. Column (1) of **Table 7** presents the test result for financing constraints, which indicates that enterprise digital transformation has a significant effect on easing financing constraints. Column (3) of **Table 7** presents the test result for internal control, which indicates a significant improvement in the quality of internal control as a result of the digital transformation of enterprises. Column (5) of **Table 7** presents the test results for

**Table 7. The mechanism test results.**

| VARIABLES | (1) FC | (2) Liquidity | (3) IC | (4) Liquidity | (5) ID | (6) Liquidity |
|---|---|---|---|---|---|---|
| DIG | -0.00096*** | 0.0330*** | 0.00641** | 0.0359*** | -0.0023*** | 0.0287** |
| | (-3.464) | (2.717) | (2.109) | (2.935) | (-3.803) | (2.443) |
| FC | | -3.815*** | | | | |
| | | (-5.478) | | | | |
| IC | | | | 1.224*** | | |
| | | | | (4.598) | | |
| ID | | | | | | -3.562*** |
| | | | | | | (-16.70) |
| Controls | Yes | Yes | Yes | Yes | Yes | Yes |
| Firm | Yes | Yes | Yes | Yes | Yes | Yes |
| Year | Yes | Yes | Yes | Yes | Yes | Yes |
| Observations | 26,526 | 26,526 | 26,526 | 26,526 | 26,526 | 26,526 |
| R-squared | 0.989 | 0.415 | 0.142 | 0.414 | 0.598 | 0.423 |

information disclosure, which indicate a significant improvement in the level of information disclosure following the digital transformation of enterprises.

The third step in empirical testing is to validate the correlation among enterprise digital transformation, mediating variables, and stock liquidity. The empirical test results are displayed in columns (2), (4), and (6) of Table 7, wherein: the test results for financing constraints are displayed in Column (2) of Table 7, DIG and FC coefficients are both significant at a 1% confidence level, suggesting that financing constraints has an intermediary effect, indicating that the mechanism by which enterprise digital transformation can alleviate financing constraints and improve stock liquidity is established. Therefore, H2 is valid. As for the test results of internal control, the coefficients of DIG and IC in Column (4) of Table 7 are both significant at a 1% confidence level. This infers that internal control plays a mediating effect, indicating that the mechanism by which enterprise digital transformation can improve the quality of internal control and thus enhance stock liquidity is established; that is, H3 is valid. As for the test results of information disclosure, the coefficients of DIG and ID in Column (6) of Table 7 are both significant at a 1% confidence level. This implies that information disclosure has an intermediary effect, indicating that the mechanism by which enterprise digital transformation can improve the information disclosure level, thus enhancing stock liquidity, is established. Therefore, H4 is valid.

# 6 Heterogeneity analysis

Based on the differences in fintech level, market development degree, and policy effect of enterprises, this study examines digital transformation's impact on stock liquidity.

## 6.1 Financial technology level

Financial technology's rapid transformation is progressively altering the ecosystem and operating environment of the capital market in which enterprises are located. This change optimizes the information mechanism and market transaction mechanism, affecting the efficacy of resource allocation and the financial service model of the financial market [68]. First, the new changes brought about by the development of fintech precisely meet the needs of the digital transformation of enterprises. First of all, the supply chain finance and trade finance generated

**Table 8. Heterogeneity analysis results.**

|  | (1) | (2) | (3) | (4) | (5) | (6) |
|---|---|---|---|---|---|---|
|  | High fintech | Medium and low fintech | Developed financial market | Backward financial market | After policy guidance | Before policy guidance |
| VARIABLES | Liquidity | Liquidity | Liquidity | Liquidity | Liquidity | Liquidity |
| DIG | 0.0326* | 0.00374 | 0.0505*** | 0.0383** | 0.0374** | 0.00415 |
|  | (1.770) | (0.197) | (2.754) | (2.371) | (2.502) | (0.151) |
| Controls | Yes | Yes | Yes | Yes | Yes | Yes |
| Firm | Yes | Yes | Yes | Yes | Yes | Yes |
| Year | Yes | Yes | Yes | Yes | Yes | Yes |
| Observations | 13,134 | 16,731 | 14,222 | 15,643 | 20,782 | 9,083 |
| R-squared | 0.177 | 0.469 | 0.411 | 0.401 | 0.261 | 0.646 |

by the development of fintech provide multi-level and efficient financing channels for enterprise financing, and optimize the mismatch between financial resources and the capital needs of enterprises in digital transformation [69]. Second, financial development of science and technology to promote the enterprise digital transformation of financial stability, financial technology to help enterprises to improve their management mechanism and financial system, reduce the efficiency financing behavior and improve the efficiency of resource allocation, digital transformation infrastructure investment and long-term maintenance, create a good enterprise digital transformation [70]. Finally, fintech helps to activate the innovation vitality in the digital transformation of enterprises, help enterprises to accurately identify the innovation prospects, reduce adverse selection and moral hazard [71], guide enterprises to pay more attention to the improvement of their core competitiveness, and promote enterprise research and development innovation to promote the digital transformation of enterprises. Therefore, the heterogeneity of digital transformation's impact on stock liquidity under different levels of fintech is tested based on regional fintech environment differences. This study draws on Li et al. (2020) [72] and adopts the news retrieval method to measure the development level of fintech, selecting 48 fintech keywords such as blockchain and digital currency to match the regions in Baidu News, adding up the number of retrieved results and taking the natural logarithm as the index of the development level of financial finance. This study performed a subgroup test according to the average level of financial technology in the region where the sample companies are located. If the level of financial technology in the company's region is higher than the average, the company is categorized as high financial technology; otherwise, it is classified as low financial technology. Columns (1) and (2) of **Table 8** present the test results, and the p-value of the chow test between the two groups was 0.027, the coefficients differ significantly between the two groups. The results show that enterprise digital transformation in high-fintech areas significantly improves stock liquidity. In contrast, the improvement of enterprises in low- and medium-fintech areas is not significant. The test results indicate that in high-fintech areas, enterprises are likelier to release the economic effect of digital transformation, enhance the information efficiency of the enterprise capital market, minimize information asymmetry, and improve stock liquidity.

## 6.2 Degree of financial market development

The primary function of the financial market is to provide capital supply and financial services to the market. Developed financial markets can offer an excellent external environment for the enterprise capital market. There are apparent differences in the quantity and quality of financial intermediaries, talents, funds, and supervision enterprises provide at different financial

market development levels [73]. The development of the financial market provides support for the effectiveness of serving the real economy, thus supporting the digital transformation of enterprises. First of all, the improvement of financial marketization drives the speed of information circulation, reduces the market information asymmetry, and helps enterprises to reduce the cost allocation and resource consumption, so as to promote the digital transformation of enterprises [74]. Secondly, the higher the degree of financial marketization, the more fierce the competition among various subjects in the market. Under the guidance of strong market competition, enterprises will focus more in resource allocation to technological innovation led by digital technology, and promote the digital transformation of enterprises [75]. Finally, the development of financial market reduces the financing cost of enterprises, improves the risk bearing capacity and risk sensitivity, promotes enterprises to implement long-term innovation activities, and promotes the digital transformation of enterprises. Therefore, based on the financial market environment of enterprises, the heterogeneity of corporate stock liquidity based on digital transformation under different financial market development levels is tested. This study draws on Fan et al. (2011) [76] to measure the degree of regional marketization. This study performed a subgroup test according to the enterprise's region's average level of financial market development. If the level of financial market development in the region where the enterprise is located exceeds the average, the enterprise is categorized as having a developed financial market; if it falls below, it is considered to have a backward financial market. Columns (3) and (4) of **Table** 8 present the test results, and the p-value of the chow test between the two groups was 0.016, the coefficients differ significantly between the two groups. The results show that enterprise digital transformation in developed financial market regions has a more pronounced effect on improving stock liquidity than in less developed financial market regions. The test results imply that the developed financial market can generate more stakeholder participation, enhance the capital market's operational efficiency, provide enterprises with more financing channels and lower financing costs, boost investors' attention and confidence, and foster the improvement of stock liquidity.

## 6.3 Digital policy guidance

Government policy guidance is vital in social and economic development and capital market operations. Digital policies propel the rapid growth of the digital economy, expand the depth and breadth of the application of digital technologies by enterprises, improve communication efficiency between enterprises and investors, boost trust among stakeholders, minimize information asymmetry, broaden financing channels, and amplify cost-saving effects. Digital policies also increase the efficiency of capital market operations, strengthen investors' market expectations, and enhance the frequency and efficiency of investors' stock trading. Therefore, based on the digital policy environment of enterprises and considering the Action Outline for Promoting the Development of Big Data issued by the Chinese government in 2015 as policy guidance, the heterogeneity of digital transformation's impact on the stock liquidity of enterprises under different policy environments is tested. Therefore, during the research period, this study conducted a subgroup test, taking 2015 as the standard. It divided the research period from 2012 to 2015 into the pre-policy guidance period and the post-policy guidance period from 2016 to 2021. Columns (5) and (6) of **Table** 8 present the test results. The results indicate that after implementing the Action Plan for Promoting the Development of Big Data, enterprise digital transformation has a significant effect on improving stock liquidity. Notably, the improvement effect is not significant before the implementation, and the p-value of the chow test between the two groups was 0.068, the coefficients differ significantly between the two groups. The test results reveal that the government provides policy guidance to bring

digital technology and resources to enterprises, amplify the digital advantages of enterprises, enhance signal transmission, minimize the adverse selection risk of investors and capital conversion losses, and thus, more effectively improve stock liquidity.

## 7 Further analysis

Digital transformation has aided companies in boosting stock liquidity. What economic effects were generated subsequent to the stock liquidity enhancement effect? The stock liquidity's impact on the capital market is reflected in two ways. First, stock liquidity can effectively minimize the risk of stock price collapse. The higher the stock's liquidity, the higher the possibility of profit of the stock. This implies higher supervision enthusiasm by stakeholders, a reduction in the short-term risk of the management, and an increase in stock prices. Simultaneously, improving stock liquidity impedes the impact of investor trading on stock prices, enhances the content of stock price information and the quality of information disclosure, reduces abnormal volatility in stock price [77], and stabilizes stock prices, thus decreasing the risk of stock price collapse. Second, stock liquidity improves the quality of analysts' forecasts. The quality of analyst prediction is influenced by the external information environment and stock information characteristics [78], while stock liquidity amplifies the efficacy of corporate information transmission. Improved stock liquidity attracts information supervision by regulators, guarantees the quality of information disclosure, and strengthens media and investors' willingness to mine information. Frequent trading by investors also accelerates information transmission efficiency, expands the channels for analysts to obtain stock price and corporate characteristic information, and effectively enhances the quality of analysts' forecasts [79,80].

Drawing from the preceding analysis, this study employs a stepwise regression technique to formulate a model that examines the economic implications of augmenting stock liquidity via enterprise digital transformation.

$$\mathbf{Con_{i,t}} = \mathbf{d_0} + \mathbf{d_1 DIG_{i,t}} + \mathbf{d_2 Controls_{i,t}} + \sum \mathbf{Firm} + \sum \mathbf{Year} + \boldsymbol{\varepsilon_{i,t}} \tag{7}$$

$$\mathbf{Con_{i,t}} = \mathbf{e_0} + \mathbf{e_1 DIG_{i,t}} + \mathbf{e_2 Liquidity_{i,t}} + \mathbf{e_3 Controls_{i,t}} + \sum \mathbf{Firm} + \sum \mathbf{Year} + \boldsymbol{\varepsilon_{i,t}} \tag{8}$$

$\mathbf{Con_{i,t}}$ stands for economic consequences, including stock price crash risk (Spcr) and Analyst Forecast Quality (Afq). The initial phase of the causal steps approach involves investigating the correlation between corporate digital transformation and stock liquidity. The positive correlation between the two has been verified in previous studies, therefore, it will not be retested in this section. Referring to Jung et al. (2023), the NCSKEW index was used to measure the stock price crash risk to test the economic effect. Additionally, we referred to Hope's (2003) test method to measure the quality of analysts' forecasts. **Table 9** presents the test results. Columns (1) and (2) of **Table 9** indicate that the estimated coefficient of enterprise digital transformation on stock price crash risk is significant at a 1% confidence level; the estimated coefficient of stock liquidity on stock price crash risk is also significant at a 1% confidence level. This indicates that stock liquidity has an intermediary effect between enterprise digital transformation and stock crash risk. It implies that enterprise digital transformation can decrease stock crash risk by improving stock liquidity. Columns (3) and (4) of **Table 9** indicate that enterprise digital transformation has a significant effect on analyst forecast quality at a 10% confidence level; the estimated coefficient of stock liquidity on analyst forecast quality is significant at a 1% confidence level. This indicates that stock liquidity has an intermediary effect between enterprise digital transformation and analyst forecast quality and implies that enterprise digital transformation can enhance the quality of analysts' forecasts by improving

**Table 9. Results of the economic consequences test.**

| VARIABLES | (1) Spcr | (2) Spcr | (3) Afq | (4) Afq |
|---|---|---|---|---|
| DIG | -0.00625*** | -0.00601*** | -0.0290* | -0.0259* |
|  | (-3.163) | (-3.045) | (-1.918) | (-1.707) |
| Liquidity |  | -0.00536*** |  | -0.0708*** |
|  |  | (-3.964) |  | (-6.096) |
| Controls | Yes | Yes | Yes | Yes |
| Firm | Yes | Yes | Yes | Yes |
| Year | Yes | Yes | Yes | Yes |
| Observations | 23,497 | 23,497 | 23,497 | 23,497 |
| R-squared | 0.041 | 0.042 | 0.061 | 0.063 |

stock liquidity. In sum, after improving stock liquidity, digital transformation can help enterprises minimize the risk of stock price crashes and improve the quality of analysts' forecasts.

We consider that there might be endogenous interference in the study regarding the economic consequences of enterprise digital transformation on stock price crash risk and analyst forecast quality, which could be resulted from reverse causality, omitted variables, and other issues. The instrumental variable method can address four types of endogenous problems, including reverse causality, omitted variables, sample and measurement errors, to a certain extent. In the study, we are not aware of the causes of endogenous problems, hence we employ the instrumental variable approach to tackle the endogenous issues arising from enterprise digital transformation that impact stock price crash risk and analyst forecast quality. This section utilizes the average value of enterprise digitalization across industries and regions as the instrumental variable (DIG.IV). The test results are presented in **Table 10**. The first stage regression test confirms the correlation between the instrumental variable and the endogenous variable. The test results are displayed in column (1) of **Table 10**. The estimated coefficient of DIG.IV is positively significant at the 1% confidence level, indicating that DIG.IV can serve as an instrumental variable. The second stage regression tests the relationship between the instrumental variable and stock price crash risk, as well as analyst forecast quality. The test results are displayed in columns (2) and (3) of **Table 10**. Among them, column (2) serves as an endogenous test for stock price crash risk, the Kleibergen-Paap rk LM statistic was significant at a

**Table 10. Endogeneity test for further analysis.**

| VARIABLES | (1) DIG | (2) Spcr | (3) Afq |
|---|---|---|---|
| DIG.IV | 0.930*** |  |  |
|  | (48.69) |  |  |
| DIG |  | -0.002*** | -0.002*** |
|  |  | (-2.241) | (-3.163) |
| Kleibergen-Paap rk LM statistic |  | 1259.595*** | 1329.969*** |
| Kleibergen-Paap rk Wald F statistic |  | 2058.616[16.38] | 2371.021[16.38] |
| Controls | Yes | Yes | Yes |
| Firm | Yes | Yes | Yes |
| Year | Yes | Yes | Yes |
| Observations | 23,497 | 23,497 | 23,497 |
| R-squared | 0.535 | 0.040 | 0.065 |

1% confidence level, thus rejecting the null hypothesis for insufficient identification of instrumental variables. Kleibergen–Paap rk Wald F statistic was much larger than the Stock-Yogo critical value 16.38. The critical value of the F test at the 10% significance level rejects the null hypothesis of the weak instrumental variable. The results indicate the absence of endogenous interference in the empirical examination of the influence of enterprise digital transformation on stock price crash risk. Column (3) represents an endogenous test of the analyst forecast quality, the Kleibergen-Paap rk LM statistic was significant at a 1% confidence level, thus rejecting the null hypothesis for insufficient identification of instrumental variables. Kleibergen–Paap rk Wald F statistic was much larger than the Stock-Yogo critical value 16.38. The critical value of the F test at the 10% significance level rejects the null hypothesis of the weak instrumental variable.The results indicate the absence of endogenous interference in the empirical examination of the influence of enterprise digital transformation on analyst forecast quality.

## 8 Conclusions and recommendations

With the accelerated growth of the digital economy, enterprise digital transformation has become the focus of scholars worldwide. To further analyze the enterprise digital transformation and its economic effects, this study examines the impact, mechanism, and economic consequences of enterprise digital transformation on stock liquidity. This study examines the sample data of Chinese A-share listed companies from 2012–2021. The analysis results are as follows:

Firstly, the digital transformation of enterprises can significantly enhance stock liquidity. This conclusion remains valid even after undergoing multiple robustness tests and endogeneity tests, indicating that enterprises in various countries have the potential to improve stock liquidity through digital transformation. This, in turn, can expand the functions of enterprise price discovery and resource allocation, thereby supporting the development of capital markets and the stability of the real economy in different countries. The research conclusion validated the influence of enterprise digitalization on the capital market in terms of stock liquidity, thereby enriching the study on the external impacts of enterprise digital transformation and the internal factors that influence stock liquidity.

Secondly, the digital transformation of enterprises has the potential to enhance stock liquidity by alleviating financing constraints, improving the quality of internal control, and improving information disclosure through three mediating mechanisms. This study delves into the impact mechanism of enterprise digital transformation on stock liquidity, viewed from the prism of corporate governance. It confirms that enterprise digital transformation can enhance the efficiency and quality of corporate governance in three key areas: financing constraints, internal control, and information disclosure. These improvements, in turn, have a positive impact on stock liquidity. It offers a fresh, practical approach for companies worldwide to amplify the economic benefits of digital transformation, boost the resilience of stock liquidity, enhance sustainable development capabilities, and enrich research on the impact mechanism of enterprise digital transformation on the capital market.

Thirdly, the results of the heterogeneity analysis reveal that: (1) The impact of corporate digital transformation on stock liquidity varies significantly under different financial technology environments. The digital transformation of enterprises in high-level financial technology regions has a significant enhancing effect on stock liquidity, whereas the enhancing effect of the digital transformation of enterprises in low- and medium-level financial technology regions is not significant. The research findings validate that the digital transformation of enterprises, when enabled by high levels of financial technology, can expedite the impact on

stock liquidity through effective utilization of resource allocation and market trading mechanisms. (2) The impact of enterprise digital transformation on stock liquidity is significant across various levels of financial market development; however, the promotion effect of enterprise digital transformation on stock liquidity is more pronounced in established financial markets. The research conclusion confirms that the digital transformation of enterprises can, to a certain extent, compensate for the drawbacks caused by lagging financial markets. It also confirms that the financing services provided by advanced financial markets can enhance stock liquidity. (3) There is a noteworthy variation in the influence of corporate digital transformation on stock liquidity prior to and following government policy guidance. After the government implemented policy guidance, the promotion effect of corporate digital transformation on stock liquidity became evident. However, prior to policy guidance, the promotion effect of corporate digital transformation was not significant. The research findings validate that enterprise's digital advantages can be expanded and their stock liquidity can be enhanced through government policy guidance and financial support.

Fourthly, the study discovered that the digital transformation of enterprises has the potential to significantly diminish the likelihood of a stock price crash and enhance the accuracy of analysts' forecasts, especially following the enhancement of stock liquidity. This section delves deeper into the economic repercussions of corporate digital transformation on external capital markets, broadening the existing research on its effect on stock price crash risk and analyst forecast quality. (1) The research conclusion verifies that the digital transformation of enterprises, with stock liquidity as an intermediary mechanism, can mitigate the impact of external factors on stock prices and reduce management short-term choices, thereby reducing the risk of stock price collapse. (2) The research conclusion verifies that the digital transformation of enterprises can effectively improve the quality of analysts' forecasts by enhancing stock liquidity, improving the willingness of external subjects such as media and investors to mine information, accelerating the efficiency of information transmission, and improving the quality and channels of obtaining analysts' characteristic information.

The digital transformation of enterprises, as a globally recognized organizational strategy, has been actively promoted and practiced in various countries. Although this study focuses on the impact of digital transformation in enterprises within the Chinese capital market, we believe that the research findings presented in this study are also relevant to capital markets in other countries. To enhance stock liquidity in the capital market and encourage the digital transformation of enterprises across different countries, this study puts forth the following suggestions:

Companies should pay attention to the impact mechanism of digital transformation on stock liquidity, recognizing the mediating roles of financing constraints, internal control, and information disclosure. Therefore, (1) Enterprises in various countries should make full use of the signaling role of digital transformation to enhance market recognition and commercial credit, broaden corporate financing channels to obtain financial support from governments, financial institutions, etc., thereby alleviating financing constraints, enhancing investment value, and reducing investment risks, in order to achieve the goal of enhancing stock liquidity [81]. Governments and institutions should fully understand the actual situation of digital transformation of domestic enterprises, formulate policies that are conducive to enterprise development, reduce administrative supervision constraints, provide subsidies for digital innovation and tax relief policies for enterprises, and act as credit guarantees for enterprises to help them expand financing channels [82]. This will offer guarantees for the digital transformation of enterprises and contribute to enhancing stock liquidity. (2) Enterprises from different countries should utilize the revolutionary impact of digital transformation on organizational processes, employing digital technology to establish effective internal control mechanisms and

organizational management models. They should strive to enhance the quality and efficiency of enterprise management, encourage external stakeholders to participate in governance supervision in order to alleviate agency conflicts, improve the effectiveness of internal control, stimulate stock market activity, and ultimately achieve the objective of improving stock liquidity [83]. Governments should adapt to the development of enterprises in a timely manner, formulate basic requirements and general standards for the construction of internal control, guide the digital transformation and internal process reform of enterprises, such as formulating uniform standards for internal capital management, procurement and sales, supply chain management, and other basic internal control activities [84], improve the ability of enterprises to create value, avoid short-term behavior of management to protect the interests of investors, reduce stock price fluctuations and thus enhance stock liquidity. (3) Enterprises from different countries must promote digital transformation in order to improve the quality of information disclosure, enhance the correlation between internal disclosure and economic returns, ensure that the disclosed information effectively reflects the intrinsic value of the enterprise, reduce investors' valuation risks and adverse selection costs, compensate to some extent for investors' information disadvantage, and increase investors' attention and trading willingness, ultimately enhancing stock liquidity [85]. Governments and regulatory agencies should formulate timely information disclosure policies suitable for their countries, expand the scope of disclosure of corporate stock price characteristic information, increase market information transparency, and supervisory agencies should enhance their inspection methods and frequency, strengthen the punishment for misconduct in information disclosure of listed companies, ensure the quality of information disclosure, boost market investor confidence, and contribute to improving stock liquidity [86]. (4) The development of enterprise digital transformation also has a new impact on policy makers and practitioners. As far as policy makers are concerned, the confidence in the digital transformation, and on the basis of balancing social and economic development, tax reduction and financial subsidies to reduce the policy makers to improve their digital technology ability to lay the foundation for serving and guiding the digital transformation strategy. For the practitioners. Enterprise managers should reduce the management's short-sighted behavior and self-interest motivation, improve the efficiency of labor input, promote enterprises to carry out digital transformation, and maximize the interests of enterprises and personal interests. Institutional investors should make use of the information "exposure effect" brought by the digital transformation of enterprises to improve the quantity and quality of information owned, reduce the proportion of follow trading and trading risks, and increase the quality of market trading, so as to curb the herd effect of stock trading.

To achieve improved performance in the capital market, enterprises should fully leverage the advantages of financial technology, financial markets, and government policies in their location, unleash the market impact of digital transformation, and assist enterprises in enhancing stock liquidity. Therefore, (1) governments should take steps to strengthen the construction of regional digital and financial technology infrastructure, rectify the imbalance of financial market resources through innovation in financial technology, enhance the accessibility of enterprise resources to alleviate resource pressure, and assist enterprises in expanding the scope of influence of the digital economy. (2) Governments should promote the growth and prosperity of financial markets. Standardized financial market supervision mechanisms, punishment mechanisms, and information disclosure systems facilitate the reduction of the risk associated with corporate self-interest behavior, ensuring the provision of high-quality financial services, boosting investor confidence, and creating a positive financial market environment [87]. Market supervision institutions should maintain the order of the financial market, develop financial market management standards, protect the capital security of enterprises and investors, provide investors with good market information to ensure the stable operation

of the capital market, and provide a favorable external market environment for the digital transformation of enterprises and the enhancement of stock liquidity. (3) Governments of different countries have implemented policies and provided financial subsidies to promote the development of the digital economy in line with their national conditions, reduce government regulations on enterprise management, assist enterprises in exploring diverse financing modes, support enterprises in carrying out digital transformation, and unleash the potential of the market economy [88]. The capital markets across various countries in the world maintain a certain level of connectivity [89]. Countries ought to enhance their exchanges and collaboration in digital technology, establish uniform technical standards, supervise the digital transformation efforts of enterprises across different countries, and enhance the overall performance of enterprises in the capital market. Supervisory institutions should provide timely attention to the promotion of government policy formulation, conduct special supervision of enterprise digital transformation policies, promptly identify issues in policy formulation and implementation, continuously improve enterprise digital transformation service policies, and assist enterprises in enhancing stock liquidity.

To promote stability and prosperity in financial markets, it is crucial for governments and enterprises to recognize the crucial role of digital transformation in reducing the risk of stock price collapse, and improving the quality of analyst forecasts, especially after enhancing stock liquidity. Enterprises in different countries should understand the digital economy's development trend and seize the digital transformation opportunity to increase stock liquidity, drive the capital market's return rate, and boost stock value. They should also improve information disclosure and external supervision mechanisms to enhance the stock market's resilience, thereby mitigating stock price fluctuations and reducing the risk of a stock price collapse. Governments should foster a favorable information environment for analysts, enhance the information oversight mechanism, and motivate investors, media, and analysts to utilize effective information (Chen et al.,2022; Yu et al., 2023) [90]. Simultaneously, companies in different countries ought to fully utilize digital technology to enhance information transparency, boost the content and quality of unique information contained in stock prices, assist analysts in improving forecasting accuracy, stabilize financial markets, and facilitate the sustainable growth of enterprises. As companies in different countries undergo digital transformation, there is reason to believe that enhanced stock liquidity will effectively mitigatate the risk of stock price collapse and improve analysts' forecasts, thereby supporting nations in maintaining the stability of their capital markets, boosting their capital market vitality, and further advancing research on the relationship between digital transformation and capital markets in various countries.

Despite this, our research still has some limitations, however, it does offer avenues for future research. Firstly, this study merely focuses on the economic impacts of corporate digital transformation on the capital market. The research conclusion is restricted to the short-term influence of digital transformation on enterprises, which fails to provide a comprehensive understanding of the overall economic effects of digital transformation on enterprises. It ignores the research regarding the long-term impact of digital transformation on enterprises, and its influence on external stakeholders. Therefore, future research should focus on the economic effects of digital transformation on the internal performance of enterprises, and its spillover effects on external stakeholders. For instance, the influence of digital transformation on green innovation performance, supply chain resilience, and the spillover effect of digital transformation on supplier technology investment. Secondly, the testing methods employed in this study's endogenous test are not capable of completely eliminating potential reverse causality, omitted variables, and other endogenous issues. There may also be endogenous problems stemming from sample selection bias and data measurement bias. Therefore, in future

research, it is advisable to employ more endogenous testing methods to minimize endogeneity issues. For example, in our research on enterprise digital transformation, we employ the Heckman two-stage model to address endogenetic issues resulting from missing or unobservable variables due to sample selection bias. We employ the GMM estimation model to introduce lagged items of two or more periods as instrumental variables in our study on enterprise digital transformation for empirical testing, thereby eliminating the influence of dynamic panel bias in our empirical testing.Thirdly, the text analysis method used in this study measures the digital transformation index of the enterprise may be affected by the characteristics of the enterprise and the stock liquidity. Therefore, in future research, more scientific measures of enterprise digital transformation should be explored, such as the combination of text analysis and executive interview. Meanwhile, the enterprise size, holding period, trading system and behavior deviation to more accurately judge the situation of enterprise digital transformation and stock liquidity. Finally, this study utilizes Chinese A-share listed companies from 2012 to 2021 as its research sample. The selection scope of research samples is limited to listed companies in China, and no listed companies from other countries or regions are selected as research samples. The selection scope of sample data is restrictive, which may not accurately reflect the impact of enterprise digital transformation on stock liquidity in regions other than China. Therefore, in future research on enterprise digital transformation, we should use listed companies in countries or regions with higher levels of enterprise digital transformation as research samples. For example, in the United States, Germany, Japan, and other countries, listed companies with leading digital technology can also serve as research samples to further study the issue of enterprise digital transformation, thereby making the research conclusions more universal and applicable.

## Supporting information

**S1 Appendix.**
(ZIP)

## Author Contributions

**Conceptualization:** Hui Liu, Jia Zhu, Huijie Cheng.

**Data curation:** Hui Liu, Jia Zhu, Huijie Cheng.

**Formal analysis:** Hui Liu, Jia Zhu.

**Funding acquisition:** Hui Liu.

**Investigation:** Hui Liu, Jia Zhu.

**Methodology:** Hui Liu, Jia Zhu, Huijie Cheng.

**Project administration:** Hui Liu.

**Resources:** Hui Liu, Jia Zhu, Huijie Cheng.

**Software:** Jia Zhu.

**Supervision:** Jia Zhu, Huijie Cheng.

**Validation:** Jia Zhu, Huijie Cheng.

**Visualization:** Jia Zhu.

**Writing – original draft:** Hui Liu, Jia Zhu.

**Writing – review & editing:** Hui Liu, Jia Zhu.

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
