## [Decision Letter · Decision Letter 0]

24 Sep 2023

PONE-D-23-30283Enterprise Digital Transformation’s Impact on Stock Liquidity: A Corporate Governance PerspectivePLOS ONE

Dear Dr. Zhu,

Thank you for submitting your manuscript to PLOS ONE. After careful consideration, we feel that it has merit but does not fully meet PLOS ONE’s publication criteria as it currently stands. Therefore, we invite you to submit a revised version of the manuscript that addresses the points raised during the review process.

We look forward to receiving your revised manuscript.

Kind regards,

Difang Huang

Academic Editor

PLOS ONE

Journal Requirements:

4. Please amend your authorship list in your manuscript file to include author Huijie Cheng.

Reviewers' comments:

Reviewer's Responses to Questions

**Comments to the Author**

1. Is the manuscript technically sound, and do the data support the conclusions?

Reviewer #1: Yes

Reviewer #2: Yes

2. Has the statistical analysis been performed appropriately and rigorously? 

Reviewer #1: Yes

Reviewer #2: Yes

3. Have the authors made all data underlying the findings in their manuscript fully available?

Reviewer #1: Yes

Reviewer #2: Yes

4. Is the manuscript presented in an intelligible fashion and written in standard English?

Reviewer #1: Yes

Reviewer #2: Yes

5. Review Comments to the Author

Reviewer #1: I would like to ask you to improve the literature review section of your manuscript by incorporating relevant papers. Specifically, we suggest that you consider the following papers:

- Bao and Huang (2021) "Shadow banking in a crisis: Evidence from FinTech during COVID-19" and Chen et al. (2022) "Dynamic correlation of market connectivity, risk spillover and abnormal volatility in stock price" for their insights on the impact of FinTech on financial markets and stock price dynamics.

- Huang et al. (2022) "Semiparametric Single-Index Estimation for Average Treatment Effects" for its methodological contribution to causal inference, which may be relevant to your analysis of the impact of enterprise digital transformation on stock liquidity.

- Yu et al. (2023a) "Cross-sectional uncertainty and expected stock returns" and Yu et al. (2023b) "Option-Implied Idiosyncratic Skewness and Expected Returns: Mind the Long Run" for their insights on stock return predictability, which may be relevant to your analysis of the economic consequences of enterprise digital transformation on stock liquidity.

We believe that incorporating these papers into your literature review may help to strengthen the theoretical foundation of your study and provide a more comprehensive understanding of the impact of enterprise digital transformation on stock liquidity.

Secondly, we would like to provide some comments to help improve your manuscript. Firstly, we suggest that you provide more details on the data and methodology used in your study, including the sample selection criteria, variable definitions, and econometric models. This will help readers to better understand the empirical analysis and replicate your results. Secondly, we suggest that you provide more discussion on the limitations of your study and avenues for future research. Specifically, you may consider discussing the potential endogeneity issues in your analysis and exploring the impact of enterprise digital transformation on other aspects of firm performance, such as profitability and innovation.

Reviewer #2: Thank you for submitting your manuscript titled "The Impact of Enterprise Digital Transformation on Stock Liquidity: Mechanisms and Economic Consequences" to PLOS ONE. After careful evaluation, I have decided to request a major revision of your manuscript before it can be considered for publication.

Overall, the study presents an interesting and important topic, exploring the impact of enterprise digital transformation on stock liquidity. However, there are several areas that need to be addressed in order to improve the clarity, rigor, and overall quality of the manuscript. I have outlined the major revisions required below:

1. Introduction:

- The introduction should provide a clear and concise overview of the research topic, highlighting the significance and relevance of studying the impact of enterprise digital transformation on stock liquidity.

- The introduction should also provide a brief literature review, discussing the existing research on this topic and identifying the research gap that your study aims to fill.

- Please ensure that the research objectives and research questions are clearly stated at the end of the introduction.

2. Methodology:

- Provide a detailed description of the data sources and sample selection process. It is important to justify the choice of Chinese A-share listed companies from 2012-2021 as the dataset for this study.

- Clearly explain the variables used to measure enterprise digital transformation and stock liquidity. Provide references for the measurement methods and discuss their validity and reliability.

- Describe the statistical methods used for data analysis, including any econometric models or techniques employed. Justify the choice of these methods and discuss their appropriateness for addressing the research questions.

3. Results:

- Present the results of the analysis in a clear and organized manner. Use tables and figures to effectively summarize and present the findings.

- Clearly state the main findings of the study and discuss their implications. Relate the findings back to the research objectives and research questions stated in the introduction.

- Provide a thorough discussion of the heterogeneity analysis results, highlighting the differences in the impact of enterprise digital transformation on stock liquidity in high-fintech areas, developed financial markets, and policy-guided enterprises.

4. Discussion:

- The discussion section should provide a comprehensive interpretation and analysis of the results, relating them to the existing literature and theoretical frameworks.

- Discuss the mechanisms through which enterprise digital transformation improves stock liquidity, specifically addressing the three mechanisms identified in the study: easing financing constraints, enhancing internal control quality, and improving information disclosure levels.

- Discuss the economic consequences of enterprise digital transformation on stock liquidity, focusing on the reduction of stock price crashes and the improvement of analysts' forecasts.

- Clearly state the limitations of the study and suggest avenues for future research.

5. Abstract:

- The abstract should be revised to accurately reflect the content and findings of the study. It should provide a clear and concise summary of the research objectives, methodology, results, and implications.

6. Writing style and clarity:

- The manuscript should be carefully proofread for grammar, spelling, and punctuation errors.

- Ensure that the writing is clear, concise, and well-organized. Use headings and subheadings to structure the manuscript and guide the reader through the content.

- Avoid excessive jargon and technical terms. Define any specialized terms or concepts that may be unfamiliar to a general audience.

6. PLOS authors have the option to publish the peer review history of their article (what does this mean?). If published, this will include your full peer review and any attached files.

Reviewer #1: No

Reviewer #2: No

---

## [Author Response · Author response to Decision Letter 0]

12 Oct 2023

Reviewer #1

1. I would like to ask you to improve the literature review section of your manuscript by incorporating relevant papers. Specifically, we suggest that you consider the following papers:

1.1 Bao and Huang (2021) "Shadow banking in a crisis: Evidence from FinTech during COVID-19" and Chen et al. (2022) "Dynamic correlation of market connectivity, risk spillover and abnormal volatility in stock price" for their insights on the impact of FinTech on financial markets and stock price dynamics.

The author's answer: 

We sincerely thank you for your valuable comments.We have carefully read the two articles by Bao and Huang (2021) and Chen et al. (2022) that you suggested.As you said, the conclusions of the financial technology-related research in the paper "Shadow banking in a crisis: Evidence from FinTech during COVID-19" can improve the theoretical basis for our research on financial technology-related issues in the heterogeneity test, and add references to "6.1 Financial technology level" in the revised manuscript as references;the factors affecting abnormal fluctuations in stock prices in the paper "Dynamic correlation of market connectivity, risk spillover and abnormal volatility in stock price" can strengthen the theoretical basis for our research on stock price crash risk, and add references to "7 Further analysis" in the revised manuscript as references.

1.2 Huang et al. (2022) "Semiparametric Single-Index Estimation for Average Treatment Effects" for its methodological contribution to causal inference, which may be relevant to your analysis of the impact of enterprise digital transformation on stock liquidity.

The author's answer: 

We sincerely thank you for your valuable comments.We have carefully read your suggestion on Huang et al. (2022) "Semiparametric Single-Index Estimation for Average Treatment Effects".As you said, after reading this article, we found that the PSM research results involved in this article can effectively improve the theoretical foundation of the PSM method used in our study for endogenous test, and we have added a reference to the "4.4.2 Missing variables problem" in the revised manuscript.

1.3 Yu et al. (2023a) "Cross-sectional uncertainty and expected stock returns" and Yu et al. (2023b) "Option-Implied Idiosyncratic Skewness and Expected Returns: Mind the Long Run" for their insights on stock return predictability, which may be relevant to your analysis of the economic consequences of enterprise digital transformation on stock liquidity.

The author's answer: 

We sincerely thank you for your valuable comments.We have carefully read the two papers you suggested, Yu et al. (2023a) and Yu et al. (2023b).As you said, we found through reading that the research conclusion on stock return prediction involved in the article "Cross-sectional uncertainty and expected stock returns" can strengthen the theoretical foundation of our research on further analysis of analyst forecasting related research, and added a reference to the "7 Further analysis" in the revised manuscript.

After incorporating the papers you suggested into our literature review, we found that these papers can effectively strengthen the theoretical foundation of our research and provide a more concise and comprehensive explanation of the impact of corporate digital transformation on stock liquidity.

2.we would like to provide some comments to help improve your manuscript. 

2.1 we suggest that you provide more details on the data and methodology used in your study, including the sample selection criteria, variable definitions, and econometric models. This will help readers to better understand the empirical analysis and replicate your results. 

The author's answer: 

We greatly appreciate your professional input into our research.As you noted, we have provided more detailed information on the data and methods used in our research in the revised manuscript, as you requested.

(1) We have added detailed information on the data and methods used in the study in the revised manuscript "3.1 Data and sample".We have added more information on the reasons for selecting the study sample period, the measurement tools used in the study, and the source of the study sample data.As you may be concerned, adding these details to the study can help readers better understand the reasons for selecting the sample.

(2) In the "3.2.1 Explained variable: stock liquidity" section of the revised manuscript, we have added more information about the definition and measurement process of stock liquidity variables, as well as the reasons and theoretical basis for selecting the Amihud research method as the measurement model.As you may be concerned, supplementing these contents in the study can help readers better understand and replicate the stock liquidity variable, and also strengthen the theoretical basis of the explained variable in the study.

(3) In the "3.2.2 Explanatory variable: digital transformation" section of the revised manuscript, we have added a more complete definition and measurement process for enterprise digital transformation, as well as the reasons and theoretical basis for selecting text analysis as a measurement method for enterprise digital transformation. In the appendix, we have also added a keyword library of enterprise digital transformation used in the study.As you may be aware, the inclusion of these additional details in the study will help readers better understand and replicate the variable of enterprise digital transformation, as well as strengthen the theoretical basis and logical representation of the explanatory variables in the study.

(4) In the "3.2.3 Control variables" section of the revised manuscript, we have added the basis for selecting control variables in the study. As you may be concerned, the inclusion of these details in the study will help readers better understand the reasons and basis for selecting control variables.

(5) In the revised manuscript, we have modified the reasons for setting the benchmark regression model, the theoretical basis for model construction, and the econometric tools used for empirical testing in "3.3 Model design". As you may be concerned, adding these elements to the study can help readers better understand the testing model set in the study.

(6) In the "4.3 Robustness test" section of the revised manuscript, we have added the reasons for choosing the replacement variable as the robustness test method in the study.In the "4.3.1 Replace the explained variable" section, we have added the purpose of testing the replacement of the explained variable and the theoretical basis for selecting the replacement variable.In the "4.3.2 Replace the explanatory variables" section, we have added the purpose of testing the replacement of the core explanatory variable, the definition of the replacement variable, and the theoretical basis for selecting the replacement variable.As you may be concerned, the supplementary explanation of the replacement variable involved in the robustness test in the study will help readers better understand the method used in the robustness test in this study.

(7) In the "4.4 Endogeneity test" section of the revised manuscript, we have added information on the endogenous test method.In the "4.4.1 Reverse causation problem" section, we have added information on the use of lagged core explanatory variables and the two-stage least squares method with instrumental variables to address the causes, theoretical basis, and testing objectives of the endogenous problem caused by omitted variables.In the "4.4.2 Missing variables problem" section, we have added information on the reasons, objectives, theoretical basis, references, and sentence expression of using PSM and multi-period DID methods to address the endogenous problem.As you may be concerned, the addition of relevant information on the endogenous test in the study can help readers better understand the purpose and significance of the methods used in the endogenous test in this study.

(8) In the "5. Mechanism analysis" section of the revised manuscript, we have added the theoretical basis and testing objectives of the stepwise testing method used for mechanism testing. As you have noted, the addition of relevant information in the stepwise testing method in the study can help readers better understand the model used in this study to test the mediating variables of financing constraints, internal control, and information disclosure, and also increases the theoretical basis for mechanism testing.

(9) In the "6 Heterogeneity analysis" section of the revised manuscript, we have added three criteria for grouping heterogeneity tests: fintech, market development level, and digital policy guidance.As you may be aware, the inclusion of grouping criteria in the study can help readers better understand the process and results of heterogeneity tests.

(10) In the "7 Further analysis" section of the revised manuscript, we have added theoretical foundations for two indicators: stock price crash risk and analyst forecast quality. As you may be aware, adding theoretical foundations to research can help readers better understand the reasons and methods for selecting economic consequences indicators.

2.2 we suggest that you provide more discussion on the limitations of your study and avenues for future research. Specifically, you may consider discussing the potential endogeneity issues in your analysis and exploring the impact of enterprise digital transformation on other aspects of firm performance, such as profitability and innovation.

The author's answer:

We greatly appreciate your professional advice on the limitations of our research and future research avenues.As you noted, we have added the limitations of the study and future research directions in the "8 Conclusions and recommendations" section of the revised manuscript.

(1) We believe that there are limitations in the scope of application of our research conclusions, which are limited to the economic effects of enterprise digital transformation on the capital market, and cannot reflect the full range of economic effects of enterprise digital transformation.As you have suggested, in our future research, we should focus on the economic effects of enterprise digital transformation on other aspects of enterprise performance, such as profitability and innovation performance.

(2) We believe that we cannot completely eliminate endogenous problems in our endogenous test.As you suggested, we should add more endogenous test models and methods, such as the Heckman two-stage model and GMM estimation method, in our future research.

(3) We believe that selecting Chinese listed companies from 2012 to 2021 as the research sample in this study is restrictive.As you have noted, in future research we need to add sample data from other countries or regions for research.

Reviewer #2

1. Introduction:

1.1 The introduction should provide a clear and concise overview of the research topic, highlighting the significance and relevance of studying the impact of enterprise digital transformation on stock liquidity.

The author's answer:

We greatly appreciate your professional advice on our research.As you may be aware, in order to provide a clearer and more concise overview of the research topic in our revised manuscript, we have integrated the first two paragraphs of the introduction into the first paragraph of the revised manuscript, making the research subject more clearly and concisely described.At the same time, we have revised the relevant statements about the significance and relevance of the study on the impact of enterprise digital transformation on stock liquidity in the first paragraph of the revised manuscript, making the significance and relevance of the study more prominent and clearly described, helping readers better understand the research topic of this study.

1.2 The introduction should also provide a brief literature review, discussing the existing research on this topic and identifying the research gap that your study aims to fill.

The author's answer:

We greatly appreciate your professional advice on our research.As you may be aware, we have restructured the original literature review in the first paragraph of the "1Introduction" section of our revised manuscript to highlight a concise discussion of the current research status of the research topic.At the same time, we have added a statement at the end of the third paragraph of the "1Introduction" section to describe the research gap that this study aims to fill, namely revealing the "mechanism black box" of the impact of enterprise digital transformation on stock liquidity. This fills the research gap in the micro-mechanism of the research topic, making the research purpose and research gap clear and explicit.

1.3 Please ensure that the research objectives and research questions are clearly stated at the end of the introduction.

The author's answer:

We greatly appreciate your professional advice on our research.As you may be aware, in order to ensure that the research objectives and research questions are reflected at the end of the introduction, we have revised and added relevant statements about the research objectives and research questions in the final paragraph of the "1 Introduction" section of the revised manuscript, making the presentation of the research questions and research objectives in this study more prominent and clear.

2. Methodology:

2.1 Provide a detailed description of the data sources and sample selection process. It is important to justify the choice of Chinese A-share listed companies from 2012-2021 as the dataset for this study.

The author's answer:

We greatly appreciate your professional input on our research.As you noted, we have added a description of the data sources and sample selection reasons used in this study in "3.1 Data and sample" of the revised manuscript, and have also included more detailed data descriptions and explanations in the attached files.

2.2 Clearly explain the variables used to measure enterprise digital transformation and stock liquidity. Provide references for the measurement methods and discuss their validity and reliability.

The author's answer:

We greatly appreciate your professional advice on our research.As you may be aware, we have revised the statement on the measurement of variables measuring corporate digital transformation and stock liquidity in "3.2 Variable setting and description" in the revised manuscript, added references to the measurement methods, and discussed the validity of the methods.

(1) We have added more descriptions of stock liquidity measurement methods and measurement processes in "3.2.1 Explained variable: stock liquidity" in the revised manuscript, clarifying the variable measurement methods for measuring stock liquidity.We have also added the research results of scholars Chiang and Zheng and Le to support and discuss the validity and reliability of the research results used in the study to measure stock liquidity using Amihud and Mendelson's research results.

(2) We have added the detailed process of measuring the variable of enterprise digital transformation in "3.2.2 Explanatory variable: digital transformation" in the revised manuscript, clarified the text analysis method used in enterprise digital transformation, and supported and discussed the effectiveness and reliability of using text analysis to measure enterprise digital transformation, using the research results of scholars such as Jiang and Fang as references.

2.3 Describe the statistical methods used for data analysis, including any econometric models or techniques employed. Justify the choice of these methods and discuss their appropriateness for addressing the research questions.

The author's answer:

We greatly appreciate your professional advice on our research.As you may be aware, we have made modifications to the sections involving data analysis methods and econometric models in the revised manuscript, and discussed the appropriateness of method selection for the research question.

(1) In the "3.1 Data and sample" section of the revised manuscript, we have added the main econometric tool used in this article, namely Stata17.0 software.

(2) In the "3.3 Model design" section of the revised manuscript, we provide a more detailed description of the econometric model used in this article's benchmark regression model, and discuss the purpose and applicability of using a multiple regression model as the benchmark regression model.

(3) In the "4.3 Robustness test" section of the revised manuscript, we have added a statement on the purpose and applicability of the test method used for robustness testing.Specifically, in the "4.3.1 Replace the explained variable" section, we have added the research conclusions of Ding and Hou, which verify the applicability and reliability of using the Zeros index and Roll index to replace the Amihud index to measure stock liquidity in the study to address the robustness issues of the benchmark regression model.

(4)In "4.3.2 Replace the explanatory variables" of the revised manuscript, we added the applicability and purpose of using the Wu et al. (2021) digital feature word library construction method and the digital feature word proportion measurement method to replace the Zhao (2021) enterprise digital transformation measurement method in our research, to verify the applicability and reliability of these two methods in solving the robustness problem of the benchmark regression model.

(5) We have added a discussion on the applicability of methods for addressing endogeneity issues in "4.4 Endogeneity test" in the revised manuscript.

Among them, "4.4.1 Reverse causation problem" supplements the reasons and purposes of using the lagged explanatory variables for periods 1-3 and the instrumental variable method in this study, and adds the research conclusions of Yonghong and Peng et al. to discuss the applicability and reliability of using this testing method to solve the endogenous problems that may exist in the benchmark regression model caused by reverse causation.

In "4.4.2 Missing variables problem", the purpose and reasons for using the multi-period DID model and PSM method to address the endogenous problem caused by omitted variables are added.The research results of Shipman, Ren et al. and other scholars are also added to discuss the applicability and reliability of using these two testing methods to solve the endogenous problem caused by omitted variables that may exist in the benchmark regression model.

(6) In the "5 Mechanism analysis" section of the revised manuscript, we have added more information on the mediating effect model used in this study, namely the stepwise approach, as well as research findings from scholars such as Baron and Kenny and Went et al., to support and discuss the applicability and reliability of using the causal steps approach to verify the impact mechanism of enterprise digital transformation on stock liquidity.

(7) In the "6 Heterogeneity analysis" section of the revised manuscript, we added three criteria for grouping heterogeneity tests: fintech, market development level, and digital policy guidance, which improved the applicability and reliability of using grouping tests to study subject heterogeneity.

(8) We have added the research conclusions of Chen, Yu, and other scholars in the "7 Further analysis" section of the revised manuscript to verify and discuss the applicability and reliability of using stock price crash risk and analyst forecast quality as indicators for economic consequences research.

3. Results:

3.1 Present the results of the analysis in a clear and organized manner. Use tables and figures to effectively summarize and present the findings.

The author's answer:

We greatly appreciate your professional input on our research.As you may be aware, we have restructured the presentation of our research findings in the "8 Conclusions and recommendations" section of the revised manuscript, reclassifying and ordering the research conclusions to make them clearer and more logically presented.

3.2 Clearly state the main findings of the study and discuss their implications. Relate the findings back to the research objectives and research questions stated in the introduction.

The author's answer:

We greatly appreciate your professional advice on our research.As you may be aware, we have revised the conclusions section of the "8 Conclusions and recommendations" in the revised manuscript to address the impact, mechanism, heterogeneity testing, and economic consequences of the research topic. This has strengthened the connection between the research conclusions and the objectives and research findings in the introduction, and fully discusses the economic significance and implications of the research conclusions.

3.3 Provide a thorough discussion of the heterogeneity analysis results, highlighting the differences in the impact of enterprise digital transformation on stock liquidity in high-fintech areas, developed financial markets, and policy-guided enterprises.

The author's answer:

We greatly appreciate your professional advice on our research.As you may be aware, we have revised the statement of heterogeneity analysis conclusions in the "8 Conclusions and recommendations" section of the revised manuscript. By comparing the research conclusions of the two subgroups of fintech, financial market development, and policy guidance, we highlight the differences in the economic impact of enterprise digital transformation on stock liquidity under different external environments.

4. Discussion:

4.1 The discussion section should provide a comprehensive interpretation and analysis of the results, relating them to the existing literature and theoretical frameworks.

The author's answer:

We greatly appreciate your professional advice on our research.As you have noted, we have revised the discussion section of the "8 Conclusions and recommendations" in the revised manuscript. We have re-interpreted and analyzed the research conclusions on the impact, mechanism, heterogeneity analysis, and economic consequences of enterprise digital transformation on stock liquidity from the government and enterprise levels. We have provided practical suggestions based on the research conclusions, and added the research conclusions of Li and Luo, Zhao, Agarwal, Lai, and other scholars in the interpretation process, combined with the theoretical framework of the previous article, making the discussion and suggestions on the research conclusions more practical and theoretical.

4.2 Discuss the mechanisms through which enterprise digital transformation improves stock liquidity, specifically addressing the three mechanisms identified in the study: easing financing constraints, enhancing internal control quality, and improving information disclosure levels.

The author's answer:

We greatly appreciate your professional advice on our research.As you may be aware, we have revised the mechanism discussion section in the "8 Conclusions and recommendations" section of the revised manuscript. We have made suggestions on the practical path of improving stock liquidity for enterprises' digital transformation from the perspectives of financing constraints, internal control, and information disclosure at the levels of governments and enterprises. We have also cited the research conclusions of Li and Luo, Zhao, Agarwal, and other scholars, as well as the theoretical framework presented earlier, to ensure the reliability and applicability of the practical suggestions proposed in this study.

4.3 Discuss the economic consequences of enterprise digital transformation on stock liquidity, focusing on the reduction of stock price crashes and the improvement of analysts' forecasts.

The author's answer:

We greatly appreciate your professional advice on our research.As you may have noticed, we have revised the discussion of the economic consequences in the "8 Conclusions and recommendations" section of the revised manuscript.We discuss the economic consequences of corporate digital transformation on stock liquidity from the perspectives of governments and enterprises, and specifically propose concrete measures to reduce the risk of stock price collapse and improve the quality of analyst forecasts. We introduce the research conclusions of scholars such as Chauhan and Chen and the theoretical framework described earlier, ensuring the reliability and applicability of the practical recommendations on economic consequences proposed in this study.

4.5 Clearly state the limitations of the study and suggest avenues for future research.

The author's answer:

We greatly appreciate your professional advice on the limitations of our research and future research avenues.As you noted, we have added the limitations of this study and future research directions in the "8 Conclusions and recommendations" section of the revised manuscript.

(1) We believe that there are limitations in the scope of application of our research conclusions, which are limited to the economic effects of enterprise digital transformation on the capital market, and cannot reflect the full range of economic effects of enterprise digital transformation.We propose that in future research, we should focus on the economic effects of enterprise digital transformation on other aspects of enterprise performance, such as profitability and innovation performance.

(2) We believe that we cannot fully eliminate endogenous problems in our endogenous test.We propose that we should add more endogenous testing models and methods, such as the Heckman two-stage model and the GMM estimation method, in future research.

(3) We believe that selecting Chinese listed companies from 2012 to 2021 as the research sample in this study is restrictive.We propose that in future research, we need to add sample data from other countries or regions for research.

5. Abstract:

The abstract should be revised to accurately reflect the content and findings of the study. It should provide a clear and concise summary of the research objectives, methodology, results, and implications.

The author's answer:

We greatly appreciate your professional feedback on the summary of our research.As you noted, we have revised the abstract in the revised manuscript to optimize the order of language presentation, and have re-summarized the research content and results of this study to ensure that the summary clearly and accurately reflects the research objectives, methods, and results of this study.

6. Writing style and clarity:

6.1 The manuscript should be carefully proofread for grammar, spelling, and punctuation errors.

The author's answer:

We greatly appreciate your professional feedback on the summary of our research.As you noted, we have carefully reviewed the manuscript and made corrections to grammatical, spelling, and punctuation errors that occurred during the research.

6.2 Ensure that the writing is clear, concise, and well-organized. Use headings and subheadings to structure the manuscript and guide the reader through the content.

The author's answer:

We greatly appreciate your professional feedback on the summary of our research.As you noted, we have revised some of the textual expressions in the revised manuscript and added subheadings where appropriate, such as adding a subheading for robustness testing in "4.4 Endogeneity test", making the textual expression of this study clearer and more concise, making it easier for readers to understand the content of this study.

6.3 Avoid excessive jargon and technical terms. Define any specialized terms or concepts that may be unfamiliar to a general audience.

The author's answer:

We greatly appreciate your professional comments on the summary of our research.As you may be aware, we often use colloquial expressions in our research to convey the content, and we try to choose nouns that are understandable to general readers for variable names.

---

## [Decision Letter · Decision Letter 1]

16 Oct 2023

PONE-D-23-30283R1Enterprise Digital Transformation’s Impact on Stock Liquidity: A Corporate Governance PerspectivePLOS ONE

Dear Dr. Zhu,

Thank you for submitting your manuscript to PLOS ONE. After careful consideration, we feel that it has merit but does not fully meet PLOS ONE’s publication criteria as it currently stands. Therefore, we invite you to submit a revised version of the manuscript that addresses the points raised during the review process.

We look forward to receiving your revised manuscript.

Kind regards,

Difang Huang, Ph.D.

Academic Editor

PLOS ONE

Journal Requirements:

Reviewers' comments:

Reviewer's Responses to Questions

**Comments to the Author**

1. If the authors have adequately addressed your comments raised in a previous round of review and you feel that this manuscript is now acceptable for publication, you may indicate that here to bypass the “Comments to the Author” section, enter your conflict of interest statement in the “Confidential to Editor” section, and submit your "Accept" recommendation.

Reviewer #1: All comments have been addressed

Reviewer #3: (No Response)

2. Is the manuscript technically sound, and do the data support the conclusions?

Reviewer #1: Partly

Reviewer #3: Partly

3. Has the statistical analysis been performed appropriately and rigorously? 

Reviewer #1: I Don't Know

Reviewer #3: I Don't Know

4. Have the authors made all data underlying the findings in their manuscript fully available?

Reviewer #1: No

Reviewer #3: No

5. Is the manuscript presented in an intelligible fashion and written in standard English?

Reviewer #1: No

Reviewer #3: Yes

6. Review Comments to the Author

Reviewer #1: To enhance the literature review, we suggest that you consider the following papers:

Huang, D., Li, Y., Wang, X., & Zhong, Z. (2022). Does the Federal Open Market Committee cycle affect credit risk? Financial Management, 51(1), 143–167.

Huang, D., Wang, X., & Zhong, Z. (2020). Monetary Policy Surprises and Corporate Credit Spreads. Available at SSRN 3700257.

Li, N., Chen, M., Gao, H., Huang, D., & Yang, X. (2023). Impact of lockdown and government subsidies on rural households at early COVID-19 pandemic in China. China Agricultural Economic Review, 15(1), 109–133.

Li, N., Chen, M., & Huang, D. (2022). How Do Logistics Disruptions Affect Rural Households? Evidence from COVID-19 in China. Sustainability, 15(1), 465.

Wu, B., Huang, D., & Chen, M. (2023). Estimating contagion mechanism in global equity market with time-zone effect. Financial Management, 52, 543–572.

Yu, D., Huang, D., & Chen, L. (2023). Stock return predictability and cyclical movements in valuation ratios. Journal of Empirical Finance, 72, 36–53.

Yu, D., Huang, D., Chen, L., & Li, L. (2023). Forecasting dividend growth: The role of adjusted earnings yield. Economic Modelling, 120, 106188.

Zhang, Y., Huang, D., Xiang, Z., Yang, Y., & Wang, X. (2023). Expressed Sentiment on Social Media During the COVID-19 Pandemic: Evidence from the Lockdown in Shanghai. Available at SSRN 4486863.

Zhou, Y., Huang, D., Chen, M., Wang, Y., & Yang, X. (2022). How Did Small Business Respond to Unexpected Shocks? Evidence from a Natural Experiment in China.

These papers are relevant to your study as they provide insights into the impact of financial technology on the financial market, the role of corporate governance in firm performance, and the dynamics of market connectivity and risk spillover. By incorporating these papers into your literature review, you can strengthen the theoretical foundation of your study and provide a more comprehensive understanding of the factors influencing stock liquidity in the context of enterprise digital transformation.

In addition to improving the literature review, we have the following comments to help you improve the submission:

1. Please provide a more detailed explanation of the methodology used in your study, including the data sources, sample selection criteria, and the econometric models employed. This will help readers better understand the robustness of your findings.

2. Consider discussing the potential endogeneity issues that may arise in your analysis, such as reverse causality or omitted variable bias. If possible, address these concerns by employing appropriate econometric techniques, such as instrumental variable estimation or propensity score matching.

3. Elaborate on the policy implications of your findings. How can policymakers and regulators use the insights from your study to promote enterprise digital transformation and enhance stock liquidity in the capital market?

4. Provide a more in-depth discussion of the limitations of your study and potential avenues for future research. This will help readers better understand the generalizability of your findings and identify areas where further investigation is needed.

Reviewer #3: I commend the authors for the substantive improvements made to the paper after the initial round of feedback. However, I still believe there's an opportunity to strengthen the introduction and motivation sections. It would be beneficial if the authors commenced their motivation with a foundational discourse on technology, encompassing areas like blockchain, AI, and their subsequent impact on corporate outcomes. I recommend starting this discussion by referencing the seminal work by An & Rau (2021) titled "Finance, technology and disruption. The European Journal of Finance, 27(4-5), 334-345." This paper offers a robust theoretical framework elucidating technology's influence on the corporate realm. Following this, a brief one or two sentences on the overarching advantages of technology within financial markets would be apt. The work by An et al. (2019), "Initial coin offerings and entrepreneurial finance: the role of founders’ characteristics. The Journal of Alternative Investments, 21(4), 26-40," serves as an excellent reference in this regard. To balance the narrative, it's crucial to also touch upon the potential drawbacks or challenges posed by technology in both the financial domain and the broader societal context. For this aspect, the study by An, Ding, and Lin (2023), "ChatGPT: tackling the growing carbon footprint of generative AI. Nature, 615(7953), 586-586," can be referenced as a pertinent example. By integrating these elements, the authors can effectively highlight the duality—both the promise and perils—of technological advancements.

7. PLOS authors have the option to publish the peer review history of their article (what does this mean?). If published, this will include your full peer review and any attached files.

Reviewer #1: No

Reviewer #3: No

---

## [Author Response · Author response to Decision Letter 1]

19 Oct 2023

Response to reviewers

Dear reviewer,

Thank you very much for your comients and professional advice. These opinions help to improve academic rigor of our study, we have mad ecorrected modifications on the revised manuscript. Meanwlile, We proofread and revise the text, grammar, and punctuation of the manuscript. We hope that our work can be improved again. Furthermore, we would like to show the details as follows:

Reviewer #1

1. To enhance the literature review, we suggest that you consider the following papers:

1.1 Huang, D., Li, Y., Wang, X., & Zhong, Z. (2022). Does the Federal Open Market Committee cycle affect credit risk? Financial Management, 51(1), 143–167.

The author's answer: 

We sincerely thank you for your valuable comments.We have carefully read your suggestion of "Does the Federal Open Market Committee cycle affect credit risk?", and found that the research conclusion in this article on the impact of the transmission of external information signals on stock risk premiums can enhance the theoretical foundation of our research on information disclosure. We have added this article as a reference in "2.2.3 Digital transformation, information disclosure, and stock liquidity".

1.2 Huang, D., Wang, X., & Zhong, Z. (2020). Monetary Policy Surprises and Corporate Credit Spreads. Available at SSRN 3700257.

The author's answer: 

We sincerely thank you for your valuable comments.We have carefully read your suggested article "Monetary Policy Surprises and Corporate Credit Spreads" and found that the research conclusions on the impact of monetary policy and other financial market conditions on corporate credit spreads in this article can enhance the theoretical foundation of our research on financial market-related heterogeneity analysis. We have added this article as a reference in the section "6.2 Degree of financial market development" for further reference.

1.3 Li, N., Chen, M., Gao, H., Huang, D., & Yang, X. (2023). Impact of lockdown and government subsidies on rural households at early COVID-19 pandemic in China. China Agricultural Economic Review, 15(1), 109–133.

The author's answer: 

We sincerely thank you for your valuable comments.We have carefully read your suggested "Impact of lockdown and government subsidies on rural households at early COVID-19 pandemic in China", and found that the research on government subsidies in this article can enhance the theoretical foundation of our research on government policy formulation in our conclusions and recommendations. We have added this article as a reference in the "8 Conclusions and recommendations".

1.4 Li, N., Chen, M., & Huang, D. (2022). How Do Logistics Disruptions Affect Rural Households? Evidence from COVID-19 in China. Sustainability, 15(1), 465.

The author's answer: 

We sincerely thank you for your valuable comments.We have carefully read your suggested article "How Do Logistics Disruptions Affect Rural Households? Evidence from COVID-19 in China." and found that the research conclusions involving supply chain management in this article can enhance the theoretical basis for our conclusions and recommendations for government policy formulation and enterprise supply chain management standards. We have added this article as a reference in the "8 Conclusions and recommendations" section.

1.5 Wu, B., Huang, D., & Chen, M. (2023). Estimating contagion mechanism in global equity market with time-zone effect. Financial Management, 52, 543–572.

The author's answer: 

We sincerely thank you for your valuable comments.We have carefully read your suggested "Estimating contagion mechanism in global equity market with time-zone effect" and found that this article validates the connectivity of global stock markets, which can enhance the theoretical basis for governments in our conclusions and recommendations to exchange and formulate unified policies. We will add this article as a reference in the "8 Conclusions and recommendations".

1.6 Yu, D., Huang, D., & Chen, L. (2023). Stock return predictability and cyclical movements in valuation ratios. Journal of Empirical Finance, 72, 36–53.

The author's answer: 

We sincerely thank you for your valuable comments.We have carefully read your suggested "Stock return predictability and cyclical movements in valuation ratios" and found that the research conclusions related to the predictability of stock returns in this article can enhance the theoretical foundation for our further research on the quality of analyst forecasts. We will add this article as a reference in the "7 Further analysis" section.

1.7 Yu, D., Huang, D., Chen, L., & Li, L. (2023). Forecasting dividend growth: The role of adjusted earnings yield. Economic Modelling, 120, 106188.

The author's answer: 

We sincerely thank you for your valuable comments.We have carefully read your suggested "Forecasting dividend growth: The role of adjusted earnings yield" and found that the research conclusions related to the factors affecting dividend forecasts in this article can enhance the theoretical basis for improving the quality of analysts' forecasts in our research conclusions and recommendations. We will add this article as a reference in the "8 Conclusions and recommendations" section.

1.8 Zhang, Y., Huang, D., Xiang, Z., Yang, Y., & Wang, X. (2023). Expressed Sentiment on Social Media During the COVID-19 Pandemic: Evidence from the Lockdown in Shanghai. Available at SSRN 4486863.

The author's answer: 

We sincerely thank you for your valuable comments.We have carefully read your suggested "Expressed Sentiment on Social Media During the COVID-19 Pandemic: Evidence from the Lockdown in Shanghai" and found that the research conclusions related to the impact of social media information dissemination in this article can enhance the theoretical basis for our research conclusions and recommendations for the government to improve the quality of information disclosure. We will add this article as a reference in the "8 Conclusions and recommendations" section.

1.9 Zhou, Y., Huang, D., Chen, M., Wang, Y., & Yang, X. (2022). How Did Small Business Respond to Unexpected Shocks? Evidence from a Natural Experiment in China.

The author's answer: 

We sincerely thank you for your valuable comments.We have carefully read your suggested "How Did Small Business Respond to Unexpected Shocks? Evidence from a Natural Experiment in China" and found that the research conclusions related to enterprises' response to external shocks in this article can enhance the theoretical basis for our research conclusions and suggestions for enterprises to improve their internal control quality. We have added this article as a reference in the "8 Conclusions and recommendations" section.

2. In addition to improving the literature review, we have the following comments to help you improve the submission:

2.1 Please provide a more detailed explanation of the methodology used in your study, including the data sources, sample selection criteria, and the econometric models employed. This will help readers better understand the robustness of your findings.

The author's answer: 

(1)In Revised manuscript "3.1Data and sample", we explained the sample selection criteria in more detail, as well as the reasons for selecting Chinese listed companies from 2012 to 2021 as the research sample.In the study, we also provided a more detailed description of the data sources, and explained the data sources and calculation methods in the accompanying documents.

(2)In Revised manuscript "3.3 Model design", we explained in more detail the reasons for selecting the multiple linear regression model as the main econometric model for this study, and explained the testing logic of the multiple linear regression model.

(3)In Revised manuscript "4.4.1Reverse causation problem", we explain in more detail the reasons for using instrumental variables as an endogenous test method in this study, explaining the logic and role of instrumental variables in the econometric process.

(4)In Revised manuscript "4.4.2Missing variables problem", we explained in more detail the reasons for using the PSM model to address the endogeneity issues caused by omitted variables in this study, and explained the logic and role of the PSM method in the econometric process.

(5)In Revised manuscript "4.4.2Missing variables problem", we explained in more detail the reasons for using a multi-period DID model to address the endogeneity problem caused by omitted variables in this study, and explained the logic and role of the multi-period DID model in the econometric process.

(6) In Revised manuscript "5 Mechanism analysis", we explain in more detail the reasons for using the stepwise testing method to verify the mechanism of the impact of enterprise digital transformation on stock liquidity in this study, and supplement the advantages and testing logic of the stepwise testing method in mediating tests.

2.2 Consider discussing the potential endogeneity issues that may arise in your analysis, such as reverse causality or omitted variable bias. If possible, address these concerns by employing appropriate econometric techniques, such as instrumental variable estimation or propensity score matching.

The author's answer: 

We greatly appreciate your professional advice on our research.As you have noted, we have carefully analyzed the endogenous issues of the impact of enterprise digital transformation on stock price crash risk and analyst forecast quality in accordance with what you said in Revised manuscript "7Further analysis". In our research, we found that instrumental variables have certain advantages in dealing with endogenous issues caused by omitted variables, reverse causality, sample selection, and measurement error. Therefore, we chose the instrumental variable method in Revised manuscript "7Further analysis" to deal with potential endogenous issues.

2.3 Elaborate on the policy implications of your findings. How can policymakers and regulators use the insights from your study to promote enterprise digital transformation and enhance stock liquidity in the capital market?

The author's answer: 

We greatly appreciate your professional advice on our research.As you have noted, we have reflected more deeply on the practical significance of our research findings for governments and regulatory agencies in Revised manuscript "8 Conclusions and recommendations", and have added suggestions for policy makers and regulators on practical paths to promote enterprise digital transformation and improve stock liquidity in the capital market.

2.4 Provide a more in-depth discussion of the limitations of your study and potential avenues for future research. This will help readers better understand the generalizability of your findings and identify areas where further investigation is needed.

The author's answer: 

We greatly appreciate your professional advice on our research.As you have noted, we have discussed in greater depth the research limitations, causes, and solutions of our research in Revised manuscript "8 Conclusions and recommendations" in accordance with your suggestions. We have also provided detailed supplementary information on the specific methods and implementation paths used to compensate for the limitations in future research.

Reviewer #3: 

1. I commend the authors for the substantive improvements made to the paper after the initial round of feedback. However, I still believe there's an opportunity to strengthen the introduction and motivation sections. It would be beneficial if the authors commenced their motivation with a foundational discourse on technology, encompassing areas like blockchain, AI, and their subsequent impact on corporate outcomes. I recommend starting this discussion by referencing the seminal work by An & Rau (2021) titled "Finance, technology and disruption. The European Journal of Finance, 27(4-5), 334-345."This paper offers a robust theoretical framework elucidating technology's influence on the corporate realm. Following this, a brief one or two sentences on the overarching advantages of technology within financial markets would be apt. The work by An et al. (2019), "Initial coin offerings and entrepreneurial finance: the role of founders’ characteristics. The Journal of Alternative Investments, 21(4), 26-40," serves as an excellent reference in this regard.To balance the narrative, it's crucial to also touch upon the potential drawbacks or challenges posed by technology in both the financial domain and the broader societal context. For this aspect, the study by An, Ding, and Lin (2023), "ChatGPT: tackling the growing carbon footprint of generative AI. Nature, 615(7953), 586-586," can be referenced as a pertinent example. By integrating these elements, the authors can effectively highlight the duality—both the promise and perils—of technological advancements.

The author's answer: 

We greatly appreciate your professional advice on our research.As you have noted, we have revised and supplemented the research on the impact of digital technology on enterprises and financial markets in Revised manuscript "1Introduction" in accordance with your suggestions.We first elaborated on the impact of digital technology on enterprises and financial markets, followed by a discussion of the advantages of digital technology-driven changes in financial markets, and finally analyzed the advantages and challenges of technological progress for markets and enterprises.

At the same time, we sincerely thank you for recommending the references you provided. We have carefully read the three articles by An & Rau (2021)"Finance, technology and disruption"; An et al. (2019)"Initial coin offerings and entrepreneurial finance: the role of founders’ characteristics"; An, Ding, and Lin (2023)"ChatGPT: tackling the growing carbon footprint of generative AI".We found that the research results on the impact of technology on enterprises in An & Rau (2021) can effectively strengthen the theoretical foundation of the impact of technology on enterprises;the research conclusions on the impact of technology on financial markets in An et al. (2019) can provide reference for the overwhelming advantages of technology in financial markets;and the research conclusions on the potential disadvantages or challenges of technology in An, Ding, and Lin (2023) strengthen the theoretical foundation of the advantages and challenges brought by digital technology to enterprises.Therefore, we will cite these three articles as references in the revision of Revised manuscript "1Introduction".

Modification description of references

In order to express the theoretical framework more clearly and concisely and enhance the theoretical foundation of this article, we have made modifications to the references in the revised manuscript.

1. The following references have been added:

Huang D, Li Y, Wang X, Zhong Z. Does the Federal Open Market Committee cycle affect credit risk? Financial Management. 2022;51(1):143-67. https://doi.org/10.1111/fima.12364

Huang D, Wang X, Zhong Z. Monetary Policy Surprises and Corporate Credit Spreads. Available at SSRN 3700257. 2020. http://dx.doi.org/10.2139/ssrn.3700257 

Li N, Chen M, Gao H, Huang D, Yang X. Impact of lockdown and government subsidies on rural households at early COVID-19 pandemic in China. China Agricultural Economic Review. 2023;15(1):109-33. https://doi.org/10.1108/CAER-12-2021-0239

Li N, Chen M, Huang D. How Do Logistics Disruptions Affect Rural Households? Evidence from COVID-19 in China. Sustainability. 2022;15(1):465. https://doi.org/10.3390/su15010465

Wu B, Chen M, Huang D. Estimating Contagion Mechanism in Global Equity Market with Time-Zone Effects. Available at SSRN 3491596. 2019. http://dx.doi.org/10.2139/ssrn.3491596

Yu D, Huang D, Chen L. Stock return predictability and cyclical movements in valuation ratios. Journal of Empirical Finance. 2023;72:36-53. https://doi.org/10.1016/j.jempfin.2023.02.004

Yu D, Huang D, Chen L, Li L. Forecasting dividend growth: The role of adjusted earnings yield. Economic Modelling. 2023;120:106188. https://doi.org/10.1016/j.econmod.2022.106188

Zhang Y, Huang D, Xiang Z, Yang Y, Wang X. Expressed Sentiment on Social Media During the COVID-19 Pandemic: Evidence from the Lockdown in Shanghai. Available at SSRN 4486863. 2023.http://dx.doi.org/10.2139/ssrn.4486863

Zhou Y, Huang D, Chen M, Wang Y, Yang X. How Did Small Business Respond to Unexpected Shocks? Evidence from a Natural Experiment in China. Evidence from a Natural Experiment in China (March 26, 2022). 2022. http://dx.doi.org/10.2139/ssrn.4044677

An J, Rau R. Finance, technology and disruption. The European Journal of Finance. 2021;27(4-5):334-45. https://doi.org/10.1080/1351847X.2019.1703024

An J, Duan T, Hou W, Xu X. Initial coin offerings and entrepreneurial finance: the role of founders’ characteristics. The Journal of Alternative Investments. 2019;21(4):26-40. https://doi.org/10.3905/jai.2019.1.068

An J, Ding W, Lin C. ChatGPT: tackle the growing carbon footprint of generative AI. Nature. 2023;615(7953):586-. https://doi.org/10.1038/d41586-023-00843-2

2. The following references were deleted:

Meng-tao C, Da-peng Y, Wei-qi Z, Qi-jun W. How does ESG disclosure improve stock liquidity for enterprises—Empirical evidence from China. Environmental Impact Assessment Review. 2023;98:106926. https://doi.org/10.1016/j.eiar.2022.106926

Sama LM, Stefanidis A, Casselman RM. Rethinking corporate governance in the digital economy: The role of stewardship. Business Horizons. 2022;65(5):535-46. https://doi.org/10.1016/j.bushor.2021.08.001

Fang VW, Noe TH, Tice S. Stock market liquidity and firm value. Journal of financial Economics. 2009;94(1):150-69. https://doi.org/10.1016/j.jfineco.2008.08.007

Karolyi GA, Lee K-H, Van Dijk MA. Understanding commonality in liquidity around the world. Journal of financial economics. 2012;105(1):82-112. https://doi.org/10.1016/j.jfineco.2011.12.008

Brogaard J, Li D, Xia Y. Stock liquidity and default riskJ]. Journal of Financial Economics, 2017, 124(3): 486-502. https://doi.org/10.1016/j.jfineco.2017.03.003

Li D, Shen W. Can corporate digitalization promote green innovation? The moderating roles of internal control and institutional ownership. Sustainability. 2021;13(24):13983. https://doi.org/10.3390/su132413983

Liu G, Liu B. How digital technology improves the high-quality development of enterprises and capital markets: A liquidity perspective. Finance Research Letters. 2023;53:103683. https://doi.org/10.1016/j.frl.2023.103683

Forcadell FJ, Aracil E, Úbeda F. The impact of corporate sustainability and digitalization on international banks’ performance. Global Policy. 2020;11:18-27. https://doi.org/10.1111/1758-5899.12761

He F, Qin S, Liu Y, Wu JG. CSR and idiosyncratic risk: Evidence from ESG information disclosure. Finance Research Letters. 2022;49:102936. https://doi.org/10.1016/j.frl.2022.102936

Mohammed A, de Sousa Jabbour ABL, Koh L, Hubbard N, Jabbour CJC, Al Ahmed T. The sourcing decision-making process in the era of digitalization: A new quantitative methodology. Transportation Research Part E: Logistics and Transportation Review. 2022;168:102948. https://doi.org/10.1016/j.tre.2022.102948

Sun Q, Shi Y. Will Digital Disclosure Affect Corporate Stocks?-Based on Stakeholders' Cognition. Engineering Economics. 2023;34(2):223-9. https://doi.org/10.5755/j01.ee.34.2.31514

Data Provision Statement

The data used in this study is provided in the supporting information in the form of an attachment.

---

## [Decision Letter · Decision Letter 2]

16 Jan 2024

PONE-D-23-30283R2Enterprise Digital Transformation’s Impact on Stock Liquidity: A Corporate Governance PerspectivePLOS ONE

Dear Dr. Zhu,

Thank you for submitting your manuscript to PLOS ONE. After careful consideration, we feel that it has merit but does not fully meet PLOS ONE’s publication criteria as it currently stands. Therefore, we invite you to submit a revised version of the manuscript that addresses the points raised during the review process.

We look forward to receiving your revised manuscript.

Kind regards,

Dariusz Siudak, Ph.D., DSc.

Academic Editor

PLOS ONE

Reviewers' comments:

Reviewer's Responses to Questions

**Comments to the Author**

1. If the authors have adequately addressed your comments raised in a previous round of review and you feel that this manuscript is now acceptable for publication, you may indicate that here to bypass the “Comments to the Author” section, enter your conflict of interest statement in the “Confidential to Editor” section, and submit your "Accept" recommendation.

Reviewer #1: All comments have been addressed

Reviewer #2: All comments have been addressed

Reviewer #3: All comments have been addressed

Reviewer #4: All comments have been addressed

Reviewer #5: All comments have been addressed

2. Is the manuscript technically sound, and do the data support the conclusions?

Reviewer #1: Yes

Reviewer #2: Yes

Reviewer #3: Partly

Reviewer #4: Partly

Reviewer #5: Yes

3. Has the statistical analysis been performed appropriately and rigorously? 

Reviewer #1: Yes

Reviewer #2: Yes

Reviewer #3: I Don't Know

Reviewer #4: (No Response)

Reviewer #5: Yes

4. Have the authors made all data underlying the findings in their manuscript fully available?

Reviewer #1: Yes

Reviewer #2: Yes

Reviewer #3: No

Reviewer #4: (No Response)

Reviewer #5: Yes

5. Is the manuscript presented in an intelligible fashion and written in standard English?

Reviewer #1: No

Reviewer #2: Yes

Reviewer #3: Yes

Reviewer #4: No

Reviewer #5: Yes

6. Review Comments to the Author

Reviewer #1: After careful review, I believe that your study has the potential to make a valuable contribution to the literature on the impact of digital transformation on stock liquidity. However, there are several areas that require significant improvement before the manuscript can be considered for publication:

- Please provide more information on the sample selection process, including any potential biases or limitations in the data.

- Consider discussing the implications of your findings for policymakers and practitioners, as well as any potential limitations or areas for future research.

Reviewer #2: (No Response)

Reviewer #3: The paper examines the relationship between digital financial inclusion and bank performance in China using data from 30 provinces from 2012 to 2021. The study finds that digital financial inclusion has a positive effect on bank performance, with risk-taking as a mediator variable. The impact of digital financial inclusion on bank performance varies across regions, with positive effects in the Northwest, South, North, and East regions and a slight negative effect in the Central region. The paper provides recommendations for banks and the government based on the findings.

The authors have addressed all my comments to a satisfactory level. I have no further comments.

Reviewer #4: Summary

This study examines the impact, mechanism, and economic consequences of enterprise digital transformation on stock liquidity, yielding four key findings. First, it reveals that enterprise digital transformation effectively enhances stock liquidity. Second, it identifies financing constraints, internal control quality, and information disclosure as the three channels through which enterprise digital transformation improves stock liquidity. Third, it establishes that the relationship between digital transformation and stock liquidity is more pronounced for policy-guided firms, and those in high-fintech areas and developed financial markets. Finally, it demonstrates that enterprise digital transformation can reduce the risk of stock price crashes and improve the quality of analysts' forecasts by improving stock liquidity.

While the topic is interesting and shows substantive improvements after the revision, this paper still faces several methodological issues. Detailed comments are provided below.

Main concerns

1. The bid-ask spread is a commonly used proxy for stock liquidity, as employed by Fang et al. (2014) and Brogaard et al. (2017). It is advisable to enhance the robustness of the baseline results using bid-ask spread to measure stock liquidity.

2. The DID model should be based on exogenous shocks, which are not determined by the research objectives, namely the firms in this paper. Constructing a quasi-natural experiment using enterprise digital transformation behavior may not be appropriate, as digital transformation is a company's own development strategy. For instance, policies related to digital transformation could provide an ideal setting for the DID model.

3. Model (5) is introduced to tackle the issue of omitted variables by examining the potential variation in the positive impact of the firm's introduction of digital transformation on stock liquidity across different levels of digital transformation. Regardless of the appropriateness of the DID model setting, there is an error in Model (5), and the accurate model should be as follows:

〖Liquidity〗_(i,t)=γ_0+γ_1 〖du〗_i+γ_2 〖dt〗_t+γ_3 〖DIG〗_(i,t)+γ_4 〖du〗_i*〖dt〗_t+γ_5 〖du〗_i*〖DIG〗_(i,t)+γ_6 〖dt〗_t*〖DIG〗_(i,t)+γ_7 〖dt〗_t*〖〖dt〗_t*DIG〗_(i,t)+∑▒Firm+∑▒Year+ε_(i,t)

And 〖du〗_i and 〖dt〗_t is omitted due to the firm-fixed effects and year fixed effects, so the final model is:

〖Liquidity〗_(i,t)=γ_0+γ_1 〖DIG〗_(i,t)+γ_2 〖du〗_i*〖dt〗_t+γ_3 〖du〗_i*〖DIG〗_(i,t)+γ_4 〖dt〗_t*〖DIG〗_(i,t)+γ_5 〖dt〗_t*〖〖dt〗_t*DIG〗_(i,t)+∑▒Firm+∑▒Year+ε_(i,t)

where γ_5 is the coefficient of interest.

4. In the mechanism analysis, financing constraint (FC), internal control (IC), and information disclosure (ID) are selected as the three channels. However, s given the relationship between stock liquidity and a firm's information transparency, it is reasonable to analyze channels related to information disclosure. Conversely, the relevance of financing constraint (FC) and internal control (IC) to stock liquidity may require further clarification. For example, the statement in line 156 suggests that “by expanding financing channels and optimizing capital structure and share capital structure, enterprises can enhance information transparency and market recognition and ease financing constraints, which are conducive to improving stock liquidity.”. Is there any reference supporting this perspective? I recommend that the authors emphasize how financing constraints and internal control contribute to either the enhancement or reduction of stock liquidity from the perspective of information transparency. Otherwise, it is better to expand the analysis for the channels of information disclosure.

5. In Table 7, the direct effect and indirect effect of digital transformation on stock liquidity are 0.0078 (0.00641*1.224) and 0.0359, respectively. The sum of the direct effect and indirect effect is 0.0437 (0.0078+0.0359), which is considerably higher than the total effect of 0.0367 as presented in the baseline results. Please double check the results. In addition, the percentage of indirect effect 0.0008 (0.0367-0.0359) in total effect is 2.18%, which is relatively small.

6. In the heterogeneity analysis, the moderators influencing the relationship between digital transformation and stock liquidity should be determinants of digital transformation. Kindly provide an explanation regarding how the financial technology level and degree of financial market development impact the firm’s digital transformation. Additionally, please include a detailed description of the measurement for these three moderators.

7. In the heterogeneity analysis, the coefficients of DIG on Liquidity are both significant, as shown in Table 8. It is recommended to provide the statistics of the coefficients difference of DIG for these two groups.

8. In the further analysis, this paper examines the economic consequences of enterprise digital transformation in improving stock liquidity., that is, Liquidity is considered as the channel through which DIG impacts stock price crash risk and analyst forecast quality. While this analysis is sound, the indirect effect of DIG on stock price crash risk through Liquidity is -0.00020 (-0.0367*0.00536), and accounts for 32% (-0.00019/-0.00625) of the total effect, which is relatively small.

9. This paper needs a proofread.

Others

1. In Table 1, why stock return is defined as earnings per share?

2. Explain why the sample period spans from 2012 to 2021.

3. The paper has more than just the following errors in formatting; please carefully address them:

(1) Apostrophes seem in different font format from others.

(2) There should a space between “quality” and “(Manita et al., 2020)” in line 42, “Stock liquidity” and “(Liquidity)” in line 255, “Table” and “3” in line 320, et al. Please carefully double check.

(3) In line 175, 320, 516, and 523, “stock” seems in different font format from others.

4. All variables used in the paper should be describe in detail.

Reference

[1] Brogaard, J., Li, D., & Xia, Y. (2017). Stock liquidity and default risk. Journal of Financial Economics, 124(3), 486-502.

[2] Fang, V. W., Tian, X., & Tice, S. (2014). Does stock liquidity enhance or impede firm innovation? The Journal of finance, 69(5), 2085-2125.

Reviewer #5: (No Response)

7. PLOS authors have the option to publish the peer review history of their article (what does this mean?). If published, this will include your full peer review and any attached files.

Reviewer #1: No

Reviewer #2: No

Reviewer #3: No

Reviewer #4: No

Reviewer #5: No

---

## [Author Response · Author response to Decision Letter 2]

21 Jan 2024

Response to reviewers

Dear reviewer,

Thank you very much for your comients and professional advice. These opinions help to improve academic rigor of our study, we have mad ecorrected modifications on the revised manuscript. Meanwlile, We proofread and revise the text, grammar, and punctuation of the manuscript. We hope that our work can be improved again. Furthermore, we would like to show the details as follows:

Reviewer #1

1. Please provide more information on the sample selection process, including any potential biases or limitations in the data.

The author’s answer: 

We sincerely thank you for your valuable comments.In Revised manuscript “3.1 Data and sample”,We supplement the details of the selection process, including the digital transformation of enterprises, stock liquidity and the possible bias in control variables in the process of selection and calculation.

2. Consider discussing the implications of your findings for policymakers and practitioners, as well as any potential limitations or areas for future research.

The author’s answer: 

We sincerely thank you for your valuable comments.In Revised manuscript “8 Conclusions and recommendations”,We supplement the impact of the results on policy makers and practitioners, and advise policy makers, business managers, and institutional investors in the context of the digital economy era.At the same time, we supplement the limitations of the enterprise digital transformation, the limitation of the stock liquidity and control variables affected in the selection and calculation process, and propose future research directions.

Reviewer #2: 

The author’s answer: 

We greatly appreciate your professional review of our study in such valuable time.

Reviewer #3:

The authors have addressed all my comments to a satisfactory level. I have no further comments.

The author’s answer: 

We greatly appreciate your professional review of our study in such valuable time.

Reviewer #4:

1. The bid-ask spread is a commonly used proxy for stock liquidity, as employed by Fang et al. (2014) and Brogaard et al. (2017). It is advisable to enhance the robustness of the baseline results using bid-ask spread to measure stock liquidity.

The author’s answer: 

We sincerely thank you for your valuable comments.In Revised manuscript “4.3.1Replace the explained variable”, We according to your suggestion by reading the references we to the difference between the selling price and purchase price and the ratio of stock liquidity, to replace the stock liquidity and benchmark regression test the robustness of the test results, the estimated coefficient of bid-ask spread in 1% of the confidence level is significantly negative, consistent with the benchmark regression results.

2. The DID model should be based on exogenous shocks, which are not determined by the research objectives, namely the firms in this paper. Constructing a quasi-natural experiment using enterprise digital transformation behavior may not be appropriate, as digital transformation is a company’s own development strategy. For instance, policies related to digital transformation could provide an ideal setting for the DID model.

The author’s answer: 

We greatly appreciate your professional review of our study in such valuable time. At the beginning of the study, we found through reading literature that many scholars believe that the digital transformation of enterprises is a positive performance in response to the continuous maturity of “Artificial Intelligence, Blockchain, Cloud Computing and Big Data” technology development, and it is an excellent quasi-natural experiment for enterprises to gradually promote their own digital transformation behavior in batches. DID model is selected to further overcome the endogenous problem: through Treatment Group and Control Group, the difference before and after the implementation of digital transformation strategy, effectively eliminate the internal differences between individuals and the bias caused by the time trend unrelated to the experimental group, and the “net effect” of enterprise digital transformation on stock liquidity can be obtained. Therefore, we also used this DID model to test the endogeneity in the benchmark regression. The DID model you proposed should be based on external shocks, and it is not appropriate to build a quasi-natural experiment with the digital transformation behavior of enterprises as the impact point, which also reminds us that this approach is questionable. Therefore, we deleted the did-d model in the study with a rigorous scientific research attitude.

3. Model (5) is introduced to tackle the issue of omitted variables by examining the potential variation in the positive impact of the firm’s introduction of digital transformation on stock liquidity across different levels of digital transformation. Regardless of the appropriateness of the DID model setting, there is an error in Model (5), and the accurate model should be as follows:

〖Liquidity〗_(i,t)=γ_0+γ_1 〖du〗_i+γ_2 〖dt〗_t+γ_3 〖DIG〗_(i,t)+γ_4 〖du〗_i*〖dt〗_t+γ_5 〖du〗_i*〖DIG〗_(i,t)+γ_6 〖dt〗_t*〖DIG〗_(i,t)+γ_7 〖dt〗_t*〖〖dt〗_t*DIG〗_(i,t)+∑▒Firm+∑▒Year+ε_(i,t)

And 〖du〗_i and 〖dt〗_t is omitted due to the firm-fixed effects and year fixed effects, so the final model is:〖Liquidity〗_(i,t)=γ_0+γ_1 〖DIG〗_(i,t)+γ_2 〖du〗_i*〖dt〗_t+γ_3 〖du〗_i*〖DIG〗_(i,t)+γ_4 〖dt〗_t*〖DIG〗_(i,t)+γ_5 〖dt〗_t*〖〖dt〗_t*DIG〗_(i,t)+∑▒Firm+∑▒Year+ε_(i,t)

where γ_5 is the coefficient of interest.

The author's answer: 

We greatly appreciate your professional review of our study in such valuable time, pointing out our problems in building the DID model and helping us to learn the DID model of preparation, which is very important for our future research and learning. Thank you again.

4. In the mechanism analysis, financing constraint (FC), internal control (IC), and information disclosure (ID) are selected as the three channels. However, s given the relationship between stock liquidity and a firm’s information transparency, it is reasonable to analyze channels related to information disclosure. Conversely, the relevance of financing constraint (FC) and internal control (IC) to stock liquidity may require further clarification. For example, the statement in line 156 suggests that “by expanding financing channels and optimizing capital structure and share capital structure, enterprises can enhance information transparency and market recognition and ease financing constraints, which are conducive to improving stock liquidity.”. Is there any reference supporting this perspective? I recommend that the authors emphasize how financing constraints and internal control contribute to either the enhancement or reduction of stock liquidity from the perspective of information transparency. Otherwise, it is better to expand the analysis for the channels of information disclosure.

The author’s answer: 

(1)We greatly appreciate your professional review of our study in such valuable time.We see you mentioned in the opinion of 156 lines of relevant description reference problem, in the process of research we refer to some literature expounds the paragraph, but after the manuscript found the article overall reference references and deleted some references to cause some paragraphs in the manuscript seems no literature support. In Revised manuscript “2.1 Digital transformation and stock liquidity”. We supplemented relevant references and use this as experience to avoid these problems in future studies.

(2)We greatly appreciate your professional review of our study in such valuable time.In Revised manuscript “2.2.1Digital transformation, financing constraints, and stock liquidity" and “2.2.2 Digital transformation, internal control, and stock liquidity”.In accordance with your suggestion, we supplement the discussion on financing constraints and internal control to improve or reduce stock liquidity from the perspective of information transparency, and further review the relationship between financing constraints and internal control and equity liquidity, so as to make up for the incomplete previous discussion and help readers better understand this study.

5. In Table 7, the direct effect and indirect effect of digital transformation on stock liquidity are 0.0078 (0.00641*1.224) and 0.0359, respectively. The sum of the direct effect and indirect effect is 0.0437 (0.0078+0.0359), which is considerably higher than the total effect of 0.0367 as presented in the baseline results. Please double check the results. In addition, the percentage of indirect effect 0.0008 (0.0367-0.0359) in total effect is 2.18%, which is relatively small.

The author’s answer: 

We greatly appreciate your professional review of our study in such valuable time.The sum of the direct and indirect effects of the internal control (IC) mentioned in your review opinion is higher than the total effect in the benchmark regression, and we tested the agreement between the results and the ones described in the manuscript by stata software. At the same time, we also note that the indirect effect of internal control (IC) you mentioned is relatively small. In future research, we will continue to look for the greater indirect effect between digital transformation and stock liquidity in enterprises.

6. In the heterogeneity analysis, the moderators influencing the relationship between digital transformation and stock liquidity should be determinants of digital transformation. Kindly provide an explanation regarding how the financial technology level and degree of financial market development impact the firm’s digital transformation. Additionally, please include a detailed description of the measurement for these three moderators.

The author’s answer: 

(1)We greatly appreciate your professional review of our study in such valuable time.In Revised manuscript “6.1Financial technology level”, we supplemented the impact of fintech on the digital transformation of enterprises from three aspects: financial resources, financial stability and innovation vitality. Meanwhile, we supplemented the calculation methods and references of the variable indicators of fintech.

(2)We greatly appreciate your professional review of our study in such valuable time.In Revised manuscript “6.2Degree of financial market development”, we supplement the impact of financial marketization degree on the digital transformation of enterprises from three perspectives of resource consumption, market competition and risk taking. At the same time, we supplemented the calculation methods of scalar indicators and references of financial marketization.

(3)We greatly appreciate your professional review of our study in such valuable time.In Revised manuscript “6.3Digital policy guidance",we supplement the way in which the digital policy-guided variable indicators are calculated.

7. In the heterogeneity analysis, the coefficients of on are both significant, as shown in Table 8. It is recommended to provide the statistics of the coefficients difference of for these two groups.

The author’s answer: 

We greatly appreciate your professional review of our study in such valuable time.According to your opinion, we chose Chow-test to test the difference between the DIG estimation coefficient of the two groups. The test results show that the difference in the DIG estimation coefficient of the two groups in fintech level, financial marketization degree and heterogeneity guided by digital policy is 0.027,0.016 and 0.068, respectively. The results show that the difference in dig coefficient of the three types of heterogeneity test results is significant.

8. In the further analysis, this paper examines the economic consequences of enterprise digital transformation in improving stock liquidity., that is, is considered as the channel through which impacts stock price crash risk and analyst forecast quality. While this analysis is sound, the indirect effect of on stock price crash risk through is -0.00020 (-0.0367*0.00536), and accounts for 32% (-0.00019/-0.00625) of the total effect, which is relatively small.

The author’s answer: 

We greatly appreciate your professional review of our study in such valuable time.We performed regression testing again using stata software to test the results consistent with those described in the manuscript. We also focus on the small indirect effect of DIG through Liquidity, and we will explore the greater economic consequences of indirect effect in future research, which also provides the direction and requirements for our future research. I want to thank you again.

9. This paper needs a proofread.

The author’s answer: 

We greatly appreciate your professional review of our study in such valuable time.After we received the review comments, we revised the language, format and font problems existing in the manuscript.

10. Others

10.1 In Table 1, why stock return is defined as earnings per share?

The author’s answer: 

We greatly appreciate your professional review of our study in such valuable time.We found that earnings per share can reflect the investment value of the company’s stock, can reflect the earnings per share for investors, can represent the company’s market performance, may affect the enterprise digital transformation and stock liquidity, thus choose earnings per share as a control variable, and in the variable description and rigorous determine the name of earnings per share. To avoid ambiguity we change Table 1 “stock return” to “earnings per share”, and change the variable definition to more detailed “The ratio of after-tax profit to the total number of shares”.

10.2 Explain why the sample period spans from 2012 to 2021.

The author’s answer: 

We greatly appreciate your professional review of our study in such valuable time.The concept of “digital transformation” was first proposed by IBM in 2012, in the study to ensure the continuity and adequacy of sample size during the study, according to the practice and the empirical test at home and abroad, and considering the huge impact of the novel coronavirus on Chinese enterprises since 2021, the node of the study period as 2021, the study period as 10 years, namely 2012-2021.To more clearly explain the sample study cycle, we added the relevant content in the revised manuscript “3.1Data and sample”.

10.3 The paper has more than just the following errors in formatting; please carefully address them:

(1) Apostrophes seem in different font format from others.

The author’s answer: 

We greatly appreciate your professional review of our study in such valuable time.We took the Apostrophes fonts format in the full text and unified all the different fonts.We found that there are two types of “ ’ ” and “ ' ” in the “Times New Roman” font. Most of the articles published by Plos One are “ ’ ”, so we unified the Apostrophe in the text as “ ’ ”. In the manuscript management system, the two symbol types may not be distinguished at the time of display, and their difference can be seen in the "Response to reviewers" file. Please consult.

(2) There should a space between “quality” and “(Manita et al., 2020)” in line 42, “Stock liquidity” and “(Liquidity)” in line 255, “Table” and “3” in line 320, et al. Please carefully double check. 

The author’s answer: 

We greatly appreciate your professional review of our study in such valuable time.We examined these issues and other parts of the manuscript and revised them. Also what you received is reviewing the manuscript may be our original submitted version. After the first two rounds of external review, ’“quality” and “(Manita et al., 2020)” in line 42’ has modified the text and corrected the problem. Meanwhile, similar issues arise elsewhere that we have revised in the latest revised manuscript. 

(3) In line 175, 320, 516, and 523, “stock” seems in different font format from others.

The author’s answer: 

We greatly appreciate your professional review of our study in such valuable time.We checked the full text of the manuscript and corrected all of the font formatting problems.

10.4 All variables used in the paper should be describe in detail. 

The author's answer: 

We greatly appreciate your professional review of our study in such valuable time.Following your opinion, we supplement the detailed calculation method of all control variables in Table 1, and the calculation method of all variables is described in detail in the “Description of indicators and data” file in the Supporting Information “S1 Appendix”.

Reviewer #5:

The author’s answer: 

We greatly appreciate your professional review of our study in such valuable time.

Modification description of references

In order to express the theoretical framework more clearly and concisely and enhance the theoretical foundation of this article, we have made modifications to the references in the revised manuscript.

1. The following references have been added:

Niu Y, Wang S, Wen W, Li S. Does digital transformation speed up dynamic capital structure adjustment? Evidence from China. Pacific-Basin Finance Journal. 2023;79:102016.

Tian J, Shao B. Financing constraints and information asymmetry of SMEs—The development of digital finance and financial risks of enterprises. Journal of the Knowledge Economy. 2023:1-21.

Brynjolfsson E, McElheran K. The rapid adoption of data-driven decision-making. American Economic Review. 2016;106(5):133-9.

Chen W, Cai W, Hu Y, Zhang Y, Yu Q. Gimmick or revolution: can corporate digital transformation improve accounting information quality? International Journal of Emerging Markets. 2022.

Brogaard J, Li D, Xia Y. Stock liquidity and default risk. Journal of Financial Economics. 2017;124(3):486-502.

Fang VW, Tian X, Tice S. Does stock liquidity enhance or impede firm innovation? The Journal of finance. 2014;69(5):2085-125.

Tsai C-H, Kuan-Jung P. The FinTech revolution and financial regulation: The case of online supply-chain financing. Asian Journal of Law and Society. 2017;4(1):109-32.

Alt R, Beck R, Smits MT. FinTech and the transformation of the financial industry. Springer; 2018. p. 235-43.

Demertzis M, Merler S, Wolff GB. Capital Markets Union and the fintech opportunity. Journal of financial regulation. 2018;4(1):157-65.

Li C, Yan X, Song M, Yang W. Fintech and corporate innovation: Evidence from Chinese NEEQ–listed companies. China Industrial Economics. 2020;1(8198):99999.

Love I. Financial development and financing constraints: International evidence from the structural investment model. The review of financial studies. 2003;16(3):765-91.

Zhao X, Wang Z, Deng M. Interest rate marketization, financing constraints and R&D investments: Evidence from China. Sustainability. 2019;11(8):2311.

Fan G, Wang X, Ma G. Contribution of marketization to China’s economic growth. Economic Research Journal. 2011;9(283):1997-2011.

2. The following references were deleted:

Zhao T, Yan N, Ji L. Digital transformation, life cycle and internal control effectiveness: Evidence from China. Finance Research Letters. 2023;58:104223. https://doi.org/10.1016/j.frl.2023.104223

Wen Z, Ye B. Analyses of mediating effects: the development of methods and models. Advances in psychological Science. 2014;22(5):731. https://doi.org/10.3724/SP.J.1042.2014.00731

Shipman JE, Swanquist QT, Whited RL. Propensity score matching in accounting research. The Accounting Review. 2017;92(1):213-44. https://doi.org/10.2308/accr-51449

Hua Z, Yu Y. Digital transformation and the impact of local tournament incentives: Evidence from publicly listed companies in China. Finance Research Letters. 2023;57:104204. 

Peng Y, Tao C. Can digital transformation promote enterprise performance?—From the perspective of public policy and innovation. Journal of Innovation & Knowledge. 2022;7(3):100198. https://doi.org/10.1016/j.jik.2022.100198

Jiang K, Du X, Chen Z. Firms' digitalization and stock price crash risk. International Review of Financial Analysis. 2022;82:102196. https://doi.org/10.1016/j.irfa.2022.102196

Chiang T C, Zheng D. Liquidity and stock returns: Evidence from international markets[J]. Global Finance Journal, 2015, 27: 73-97. https://doi.org/10.1016/j.gfj.2015.04.005

Yonghong L, Jie S, Ge Z, Ru Z. The impact of enterprise digital transformation on financial performance—Evidence from Mainland China manufacturing firms. Managerial and Decision Economics. 2023;44(4):2110-24. https://doi.org/10.1002/mde.3805

Ding R, Hou W. Retail investor attention and stock liquidity. Journal of international financial markets, institutions and money. 2015;37:12-26. https://doi.org/10.1016/j.intfin.2015.04.001

Roll R. A simple implicit measure of the effective bid‐ask spread in an efficient market. The Journal of finance. 1984;39(4):1127-39. https://doi.org/10.1111/j.1540-6261.1984.tb03897.x

Jung H, Song C-K. Managerial perspectives on climate change and stock price crash risk. Finance Research Letters. 2023;51:103410. https://doi.org/10.1016/j.frl.2022.103410

Chen W, Zhang L, Jiang P, Meng F, Sun Q. Can digital transformation improve the information environment of the capital market? Evidence from the analysts' prediction behaviour. Accounting & Finance. 2022;62(2):2543-78. https://doi.org/10.1111/acfi.12873

Fang M, Nie H, Shen X. Can enterprise digitization improve ESG performance? Economic Modelling. 2023;118:106101. https://doi.org/10.1016/j.econmod.2022.106101

Le H, Gregoriou A. How do you capture liquidity? A review of the literature on low‐frequency stock liquidity. Journal of Economic Surveys. 2020;34(5):1170-86.

Fang M, Nie H, Shen X. Can enterprise digitization improve ESG performance? Economic Modelling. 2023;118:106101. https://doi.org/10.1016/j.econmod.2022.106101

Hope OK. Disclosure practices, enforcement of accounting standards, and analysts' forecast accuracy: An international study. Journal of accounting research. 2003;41(2):235-72. https://doi.org/10.1111/1475-679X.00102

Data Provision Statement

The data used in this study is provided in the supporting information in the form of an attachment. The support information file name is called: S1 Appendix.

---

## [Decision Letter · Decision Letter 3]

5 Mar 2024

Enterprise Digital Transformation’s Impact on Stock Liquidity: A Corporate Governance Perspective

PONE-D-23-30283R3

Dear Dr. Zhu,

We’re pleased to inform you that your manuscript has been judged scientifically suitable for publication and will be formally accepted for publication once it meets all outstanding technical requirements.

Kind regards,

Dariusz Siudak, Ph.D., DSc.

Academic Editor

PLOS ONE

Additional Editor Comments (optional):

Reviewers' comments:

Reviewer's Responses to Questions

**Comments to the Author**

1. If the authors have adequately addressed your comments raised in a previous round of review and you feel that this manuscript is now acceptable for publication, you may indicate that here to bypass the “Comments to the Author” section, enter your conflict of interest statement in the “Confidential to Editor” section, and submit your "Accept" recommendation.

Reviewer #1: All comments have been addressed

Reviewer #4: All comments have been addressed

2. Is the manuscript technically sound, and do the data support the conclusions?

Reviewer #1: Yes

Reviewer #4: Yes

3. Has the statistical analysis been performed appropriately and rigorously? 

Reviewer #1: Yes

Reviewer #4: Yes

4. Have the authors made all data underlying the findings in their manuscript fully available?

Reviewer #1: Yes

Reviewer #4: Yes

5. Is the manuscript presented in an intelligible fashion and written in standard English?

Reviewer #1: Yes

Reviewer #4: Yes

6. Review Comments to the Author

Reviewer #1: (No Response)

Reviewer #4: (No Response)

7. PLOS authors have the option to publish the peer review history of their article (what does this mean?). If published, this will include your full peer review and any attached files.

Reviewer #1: No

Reviewer #4: No

---

## [Editor Report · Acceptance letter]

25 Oct 2023

PONE-D-23-30283R2 

Enterprise Digital Transformation’s Impact on Stock Liquidity: A Corporate Governance Perspective 

Dear Dr. Zhu:

I'm pleased to inform you that your manuscript has been deemed suitable for publication in PLOS ONE. Congratulations! Your manuscript is now with our production department. 

Kind regards, 

on behalf of

Prof. Difang Huang 

Academic Editor

PLOS ONE